

# Review Article: Leveraging Social Media for Managing Natural Hazard Disasters: A Critical Review of Data Collection Strategies and Actionable Insights

Lakshmi S. Gopal[1], Rekha Prabha[1], Hemalatha Thirugnanam[1], Maneesha Vinodini Ramesh[1], and Bruce D. Malamud[2]

[1]Center for Wireless Networks & Applications (WNA), Amrita Vishwa Vidyapeetham, Amritapuri, India
[2]Institute of Hazard, Risk and Resilience (IHRR), Durham University, Durham, DH1 3LE, UK

**Correspondence:** Lakshmi S. Gopal (lakshmisgopal@am.amrita.edu)

**Abstract.** This paper critically reviews 250 articles from 2010 to September 2023, analyzing how social media data is utilized in disaster management, addressing challenges in relevance filtering and noise reduction to extract actionable disaster information and enhance decision-making efficiency. The results of our critical analysis are given in a Social Media Literature Database where we categorize each article's information into 7 main categories and 27 subcategories, covering article details, case study regions, disaster events, social media data specifics, data collection and analysis methods, and evaluation methodologies. To assess the effectiveness of social media in providing actionable disaster information, we further classify the articles into 9 categories, covering public discourse analysis, temporal and spatial insights, relevance filtering methods, community/stakeholder collaborations, disaster trends, and resource identification. We also illuminate historical disaster events within the review period and discuss the results through graphical visualizations. Our findings show that natural language processing methods, particularly content analysis, were commonly utilized in the literature, and contribute significantly to basic data filtering by removing noise. Commonly used advanced robust analysis machine learning methods included Support Vector Machines, Naive Bayes, and Neural Networks. We found that proficiency in temporal and spatial analysis of social media data is widespread among the studies, with varying success in implementing effective relevance filtering. Our actionable information categorization revealed a need for further exploration into community interactions and resource identification using social media data during and after disasters. Based on the literature study and our own experience on the subject, we propose six best practices for social media usage in disaster situations for the community and five best practices for researchers to enhance disaster management strategies.

## 1 Introduction

In the age of information, social media has become a powerful platform for communication and rapid information dissemination (McCormick et al., 2017; Wang et al., 2018; Li et al., 2018b; Fauzi, 2023). Social media platforms introduced a new direction in assisting in disaster management, enhanced situation awareness, analysing emotions, and community interaction analysis discovering solutions unified with current technologies (Bruns and Liang, 2012; Gerlitz and Rieder, 2013; Kryvasheyeu et al.,





2016; Martínez-Rojas et al., 2018). This critical review explores the multifaceted relationship between social media and disaster management, aiming to identify gaps, provide insights, and offer potential future directions.

While traditional media sources like newspapers, television, and radio offer reliable information, social media provides distinct advantages, including convenient access to information, interactive community engagement, and diverse situational insights from various perspectives and locations (Chatfield and Brajawidagda, 2013; Dashti et al., 2014; Li et al., 2015; Stieglitz et al., 2018; Wang and Ye, 2018). However, the challenge lies in sifting through the abundance of information to identify trustworthy and pertinent data (Smith et al., 2017; Gulnerman and Karaman, 2020; Srivastava et al., 2020).

This is particularly critical in disaster scenarios where the spread of rumours and misinformation is unacceptable (Cenni et al., 2017; Yan et al., 2017). It is also important that the data extracted from social media platforms must be actionable for disaster response, recovery, relief, and rapid decision-making by authorities (Sriram et al., 2010; Li et al., 2015; Cenni et al., 2017). This critical review focuses on the process of discerning relevant and actionable data from social media to enhance disaster response and recovery efforts.

Existing literature reviews on social media data (SMD) platform evaluations, data collection tools, and analysis methods over time (Cheng et al., 2016; Shibuya and Tanaka, 2019; Kitazawa and Hale, 2021). These reviews address the utility of social media data across various phases of disaster management. However, limited attention has been devoted to the collection and analysis of topic-relevant data with an emphasis on noise reduction for method enhancement. Even when literature explores topic discovery methods (Volkova, 2014; Čišija et al., 2018; Qarabash and Qarabash, 2018), less focus is placed on assessing

the actionability of discovered data in disaster scenarios. This critical review meticulously examines the literature, aiming to establish a classification system for actionable information, thereby assessing the practical value of social media data in disaster management.

   The purpose of this critical review is twofold. First, we seek to evaluate the existing literature on the topic of social media usage for managing disasters where we discuss the key findings, and methodologies used for relevance filtering of social media data.

Second, we aim to perform an in-depth analysis of how the existing solutions aided in bringing out 'actionable information' from social media data. By performing this critical review we aim to shed light on the various methods of social media data analysis to identify pertinent data and to suggest future directions.

   Throughout the following sections, we discuss the methodologies used in the existing body of literature, major disaster events in the past decade, and emerging trends, and offer recommendations for future studies. By doing so, we hope to gain a deeper

understanding of how social media data analysis can play a relevant role in improving rapid decision-making during a disaster scenario by assisting policymakers, emergency responders, researchers, and the general community.

   The manuscript is organised as follows. In section 2 we describe the background. In Section 3 we present the critical review methodology which includes sub-sections detailing research question identification and the steps in constructing our Social Media Literature Database (Gopal et al. (2023)). In Section 4, we bring in the results of the critical review methodology. In

Section 5, we critically discuss all the categories in our Social Media Literature Database (Gopal et al. (2023)) to present insightful information and proposes best practices to utilise social media data for the community and researchers to improve disaster management strategies. Finally, in section 6, we summarise our analysis based on the lessons learned.



## 2 Background

In the last decade, computer scientists have made significant contributions to organizing and evaluating data in disaster man-
agement. In this background we focus on critical and systematic review articles within the context of social media on data
collection, analysis, relevant data identification, as well as applications, opportunities, and challenges in various phases of dis-
aster management.

In the late 2010s, a seminal work by Hristidis et al. (2010) investigated data integration, information extraction, retrieval, fil-
tering, data mining, and decision support methods in disaster management. This article, among the earlier ones (Imran et al.,
2015; Granell and Ostermann, 2016), emphasised the necessity of establishing a disaster management dataspace and outlined
associated challenges. Concurrently, Veil et al. (2011) systematically reviewed articles that addressed the creation of risk and
crisis management rules and processes, enabling community engagement in decision-making systems.

Several authors emphasise the potential of social media to enhance community interaction across all disaster management
phases (Tim et al., 2017; Anson et al., 2017; Nazer et al., 2017). A pivotal work by Landwehr and Carley (2014) extensively
explores the roles of the community and organisations in disaster management. They illuminate how the public not only seeks
life-saving information but can also contribute to effective information dissemination, fostering community awareness. Addi-
tionally, the authors critically review how first responder organisations, including government agencies, increasingly rely on
social media data to identify areas in need of assistance during crises.

In recent years, some authors conducted comprehensive reviews and discussions on advanced data acquisition, characteriza-
tion, and preparation methods (Houston et al., 2015; Spence et al., 2016; Eriksson, 2018; Zhou et al., 2018; Luna and Pennock,
2018; Saroj and Pal, 2020). They examined technical approaches, including API calls and social media querying. These articles
also emphasised the importance of data pre-processing, geolocation identification, and geocoding for effectively identifying
affected areas. One of the early articles in the literature database, Imran et al. (2015), provided detailed insights into disaster
event detection using social media data, covering the aforementioned methods.

One of the prominent challenges addressed in these review articles is the security and quality of data, particularly its relevance
in disaster management (Simon et al., 2015; Lin et al., 2016; Said et al., 2019; Acikara et al., 2023). Data quality is often
associated with data credibility, where concerns about rumours or false information arise (Reuter and Kaufhold, 2018; Jurgens
and Helsloot, 2018). A significant article authored by Haworth and Bruce (2015) discusses the risks associated with data from
untrained individuals with diverse agendas and expertise, emphasising the lack of quality assurance. The article underscores
the critical importance of accurate information in disaster scenarios, as a delay due to waiting for official notification could
have life-threatening consequences and pose a significant danger.

In their study, Nazer et al. (2017) focused on various misleading content, including rumours, spam, and bot-generated data,
emphasising the necessity of filtering such content before its utilisation in any phase of disaster management. The article also
observed that social media users' language changes due to distress. The victim's cognitive processing and social orientation
during or after a disaster lead to changes in language at the sentence or topic level. The authors argued that probabilistic meth-
ods, such as Latent Dirichlet Allocation (LDA), can be employed to extract the probability of each topic in a document.



Recent reviews have offered comprehensive insights into managing disaster scenarios using advanced AI technologies and social media data (Pender et al., 2014; Yu et al., 2018; Goswami et al., 2018; Eckert et al., 2018; Akter and Wamba, 2019; Vongkusolkit and Huang, 2021; Aboualola et al., 2023). In a pivotal work by Imran et al. (2020), the article highlights the multimodal nature of SMD, encompassing text, images, videos, relevant URLs, and additional metadata. These modalities often provide complementary information that, when collectively analysed, can significantly enhance the understanding of a crisis. The article delves into AI and ML-based solutions for effectively handling multimodal data.

Several authors conducted bibliometric analyses in the aforementioned field. In a significant article by Tang et al. (2021), it was noted that the use of social media data for disaster management achieved stability during the period from 2015 to 2019. The authors observed that advanced technologies in NLP, ML, and computer vision emerged as key focal points in disaster management. Additionally, Fauzi (2023) conducted a bibliometric analysis to analyse and predict future trends and developments in social media applications.

An important finding derived from the review articles is that social media significantly facilitates community interactions in crises (Alexander, 2014; Reuter and Kaufhold, 2018). However, the primary concern lies in the unreliability of the data, which hinders its practicality as actionable information during disasters (Beigi et al., 2016; Palen and Hughes, 2018; Zhang et al., 2019a; Imran et al., 2020).

## 3 Critical Literature Review Methodology

To construct the Social Media Literature Database (Gopal et al. (2023)), we conducted a critical review of pertinent English-language articles using 'social media' and 'disaster management' related keywords, primarily sourcing content from Google Scholar. The time period covered was from 2010 to September 2023. Section 3.2 details the specific search criteria employed in building the literature database. A two-stage screening process was implemented: an initial assessment based on titles and abstracts to shortlist relevant articles, followed by a critical review of the selected articles to confirm their relevance to the research topic.

We have taken elements from Boaz et al. (2002) to follow a specific protocol for the critical literature review:

i Focusing on answering a specific question(s)

ii Seeking to identify relevant research

iii Synthesising the research findings in the studies included

iv Aiming to be as objective as possible about research to remove bias

In this paper, we followed a critical literature review with four major stages as shown in figure 1 and each stage is described in the following sub-sections.





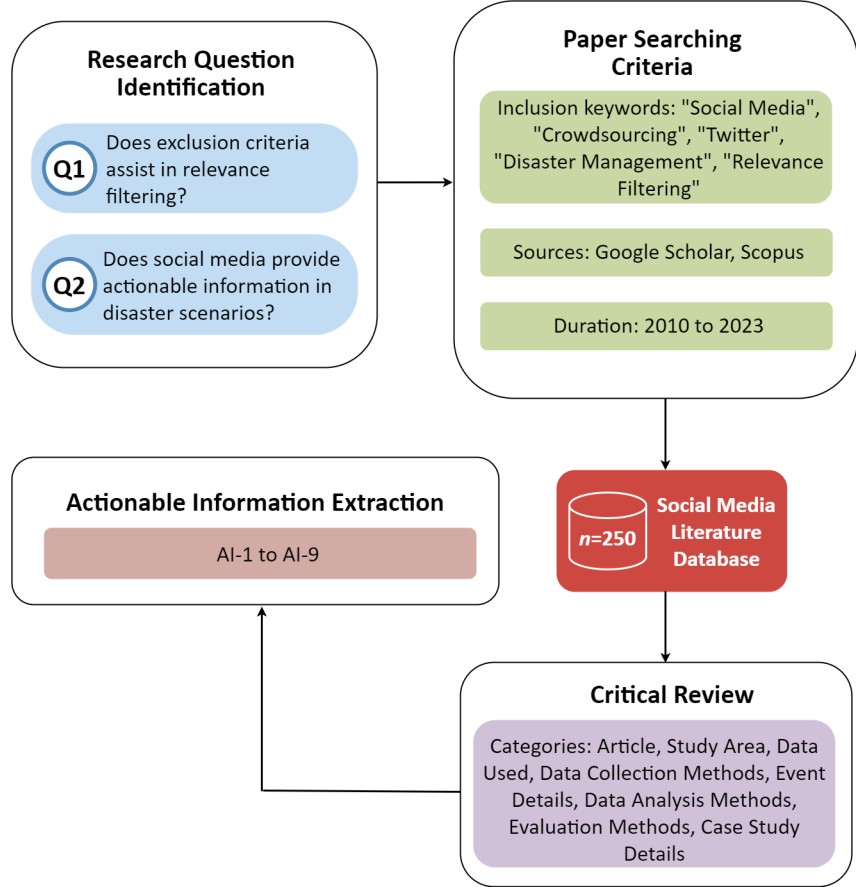

**Figure 1.** The block diagram summarises the critical review methodology with four major stages and their respective details.

## 3.1 Research Question Identification

In a hazard scenario, Volunteered Generated Information (VGI) generated through social media is advantageous, but due to lack of reliability and increased generation of data, rapid decision-making is affected (Black et al., 2012; Ashktorab et al., 2014; Radianti et al., 2016; Yan et al., 2017; Kankanamge et al., 2020b). Considering these issues, we have derived the following research questions.

– Q1: Do exclusion criteria assist in relevance filtering of social media data?

Exclusion criteria are conditions used to eliminate irrelevant data, facilitating relevance filtering. These criteria often consist of keywords or phrases related to topics to be excluded from large datasets and are considered a straightforward method for reducing irrelevant data.

We analyse articles in the literature that utilise social media data to identify the methods applied for relevance filtering. Our specific focus is on articles that have experimented with exclusion criteria in data collection. Our objective is to





determine whether Natural Language Processing (NLP) methods can effectively aid in relevance filtering for basic data collection or if Machine Learning (ML) methods are required.

– Q2: Does social media provide actionable information in disaster scenarios?

A significant drawback of social media data is its credibility (Win and Aung, 2017; Ravi Shankar et al., 2019; Nair et al., 2022; Loynes et al., 2022). Social media users encompass various categories, including public users, government organisations, NGOs, public figures, and news media. During a disaster scenario, government, non-government organisations, and news media typically provide trustworthy information about the crisis. However, public posts may also include valuable emergency information from actual victims, often in the form of photos or videos (Khaleq and Ra, 2018; Banujan

et al., 2018).

Inaccurate information may be disseminated, whether intentionally or unintentionally, including the spread of rumours or discussions about similar disaster events occurring elsewhere (Remy et al., 2013; Musaev et al., 2018; Arapostathis, 2021). This challenge underscores the difficulty in identifying relevant data that can be considered actionable. In this context, actionable information is defined as data that facilitates prompt decision-making in disaster scenarios.

We have defined various forms of actionable information from social media data, as detailed in Section 4.3. We reviewed the articles in the database to ascertain if they proposed solutions for extracting actionable information. Our objective is to gain a comprehensive understanding and determine whether social media indeed contributes to effective disaster management by providing pertinent information for rapid decision-making. By addressing these research questions, we also aim to offer optimal guidance for investigators regarding the extent to which social media contributes to disaster

management research.

To address the above research questions, we bring in 7 main categories in our critical review literature database where data related to the following questions will be placed:

(a) What are the methods opted to collect disaster-related social media data?

(b) What are the existing methods of relevance or domain filtering of social media data, within and outside disaster
scenarios?

(c) What are the methods of exclusion criteria usage for relevance filtering?

(d) Does the literature further analyse the exclusion criteria to avoid missing data and not to include irrelevant data?

(e) What are the existing data analysis methods used, specifically using ML and NLP?

(f) Does the literature address the issue of false information dissemination?

(g) What approaches have the articles introduced to identify, analyse, and extract actionable information?

## 3.2 Paper Searching Criteria

To construct the Social Media Literature Database (Gopal et al. (2023)), we searched articles in Google Scholar and Scopus online platforms using the keywords related to disaster management and data science (see figure 1) and created five Boolean





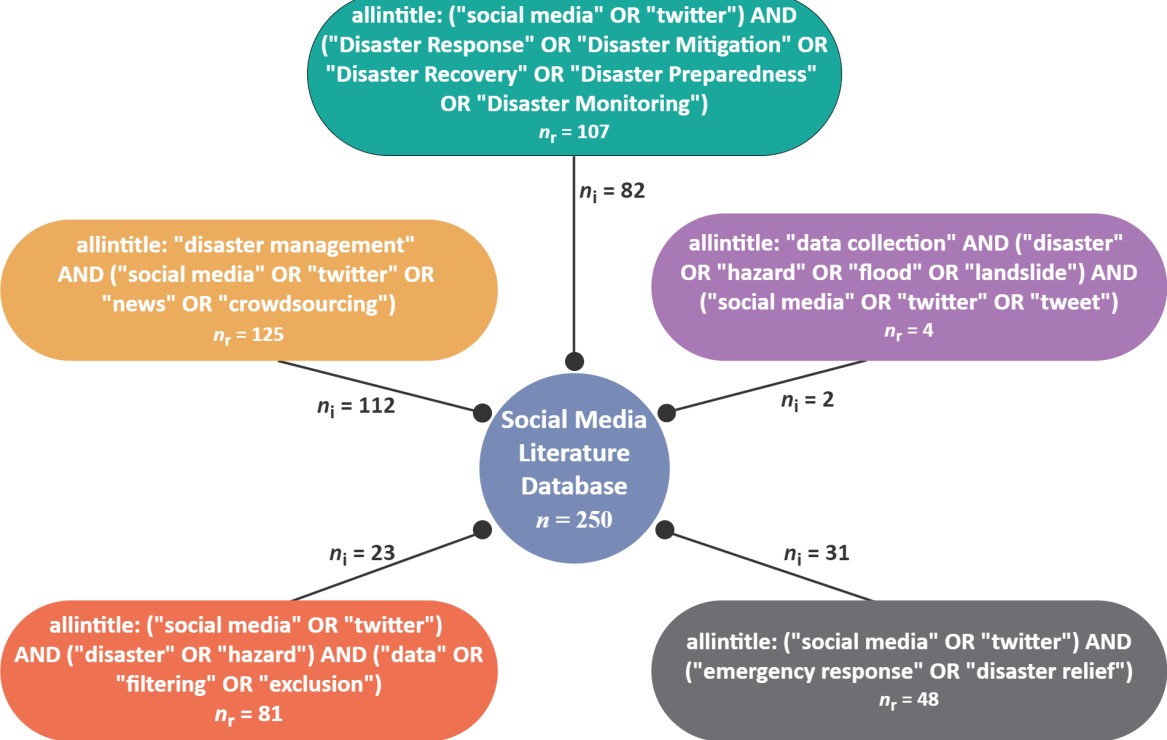

**Figure 2.** Boolean search strings used for article collection to list in the literature database - The search strings are applied on the article title for searching in Google Scholar (Last queried on September 2023). $n_r$ represents the number of resultant articles of each Boolean search string, $n_i$ represents the number of articles included in the literature database from each search string and n represents the total number of articles in the literature database.

search strings applied to the title of the articles. Figure 2 shows the Boolean search strings used to collect the articles to create

a literature database. We have used five boolean search strings and each string generated a certain number of articles, depicted as $n_r$. After an analysis of these articles, we included a certain number of articles in the literature database, depicted as $n_i$. Following the application of search terms, each resulting article was manually reviewed to determine its relevance, specifically focusing on social media, disaster management, or relevance filtering. It is important to note that we did not implement any exclusions during the collection process. In total, we have included 250 articles in our database, selecting various data types

for extraction to compile the database.

We compiled the database by considering peer-reviewed journals, conference papers, and reports. The articles included in the database span from 2010 to September 2023. The majority of these articles are associated with disaster management, with a subset related to social media data analytics and social science, where methodologies for relevance filtering using social media data are employed.

During phase I of paper selection for the critical review, we used the disaster-related keywords in the titles as described in the





boolean search strings (see figure 2), and then reading the abstract of the article for applicability. The may however be other keywords that we did not use which would have identified further relevant literature. For instance, a seminal work by Niles et al. (2019) is indexed in Google Scholar and can be found when searched with the keywords 'social media' and 'natural hazards'. However, using the developed boolean search strings, this article did not get listed in the database, as we missed it

due to the absence of relevant keywords with plural combinations.

For example, the keyword search in Google Scholar for (("Social Media") AND ("Natural Hazard" OR "Natural Hazards")) identifies 30 articles. Of these 30, after reading the abstracts, 7 would have been included in our Database, but only one actually was included. Examples of those not included are "Rapid flood inundation mapping using social media, remote sensing and topographic data" (Rosser et al., 2017), "Sub-event discovery and retrieval during natural hazards on social media data" (Wu

et al., 2016), "Detecting Natural Hazard-Related Disaster Impacts with Social Media Analytics: The Case of Australian States and Territories" (Yigitcanlar et al., 2022), and "Public Attention to Natural Hazard Warnings on Social Media in China" (Hu et al., 2019).

### 3.3 Synthesis of Research Findings

We extracted seven major categories, each containing multiple sub-categories, which were added to the database, as depicted

in Figure 3. For each of the 250 papers, we conducted reviews to identify information that could be assigned to these seven categories and their respective subcategories. In addition to this data, we also conducted an actionable information analysis of the articles, as detailed in Section 4. In the following subsections, we provide descriptions of the collected information under each category and its subcategories, along with an explanation of the data grouping procedure for each category.

#### 3.3.1 Category A: Article Description

In the Social Media Literature Database, article description (A) comprises nine unique sub-categories, as depicted in Figure 3. Subcategory A1, 'ID,' provides a unique identification number for each article, while subcategory A2, 'Title,' lists the article's title in the database. Subcategory A3, 'Author(s),' denotes the authors of each article, with a range of 1 to 4 authors per article. Subcategory A4, 'Theme,' categorises each article into one of the themes: 'Disaster Management,' 'Social Media Analytics,' or 'Social Science,' with each article assigned to a specific category.

Subcategory A5, 'Year,' indicates the publication year of the article. Subcategories A6 and A7 specify the kind and type of article, respectively. 'Kind of article' classifies an article as 'Journal,' 'Conference,' 'Report,' or 'Book,' while 'Article Type' designates an article as either 'Survey' or 'Other.' In this context, 'Survey' includes survey or review papers related to each theme, and 'Other' encompasses articles with analytical, experimental, or interpretative characteristics. Subcategory A8, 'Publication Name,' details the title of the journal, conference, or book series in which the article was published. Finally,

subcategory A9, 'Citations,' reflects the number of citations an article has accumulated as of September 2023.





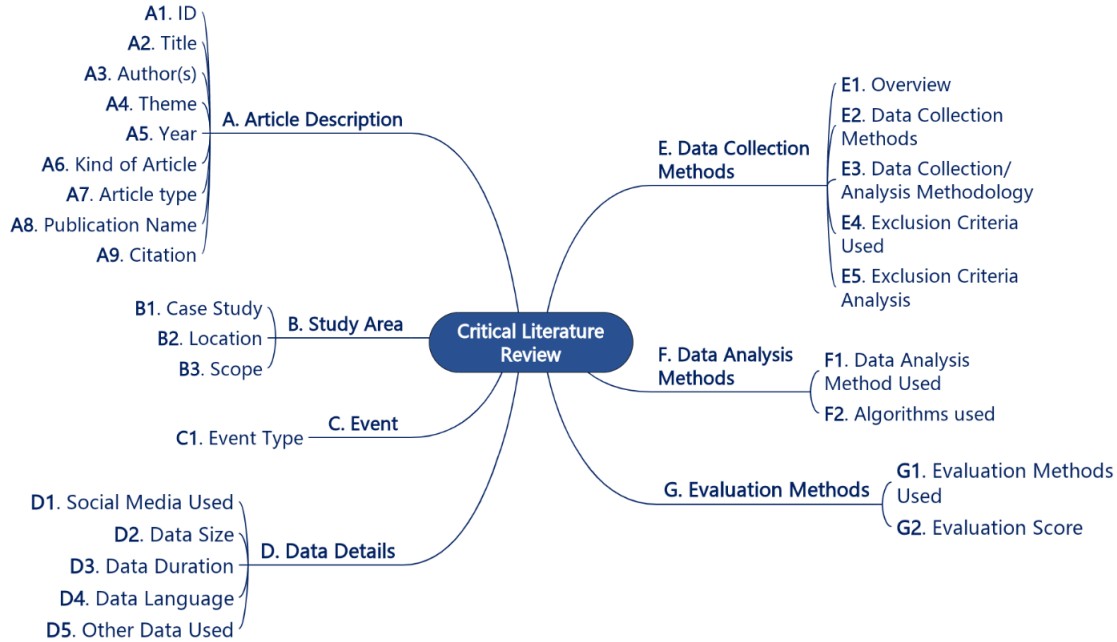

**Figure 3.** The mind map summarises the seven main categories and their respective subcategories populated in the critical review database.

### 3.3.2 Category B: Study Area

In the literature database, Category B provides study area details with three subcategories. Subcategory B1, 'Case Study,' is a binary field (yes or no) indicating whether an article employs a case study to support its methodology. If an article uses a case study, we fill subcategory B2, 'Location,' with the relevant place names in focus.

An article may incorporate case studies related to disaster events affecting areas at various spatial scales, as illustrated in Figure 4. This figure delineates the spatial scales considered in the study, along with associated terminology and their characteristics. Case studies are categorized into subcategory B3, labeled 'Scope,' which characterizes geographic scales using the terminologies 'national,' 'regional,' and 'local.' In the 'national' category, articles focus on cases where the disaster event impacts one or more countries. The 'regional' and 'local' categories group articles with case studies specific to disaster events within states,

districts, large counties, cities, and towns. Figure 4 offers an overview of data characteristics and spatial scales corresponding to each scope terminology. For categorization, 'regional' encompasses countries, states, districts, counties, and provinces within a range of a million km$^2$. A 'local' scope includes towns, cities, or villages within a range of ten thousand km$^2$.

### 3.3.3 Category C – Event

We categorise the type of disaster event in subcategory C1, 'Event Type,' which includes categories like 'Landslide,' 'Flood,'
'Earthquake,' 'Hurricane,' or other natural or man-made disaster event names. Some articles focus on disaster management



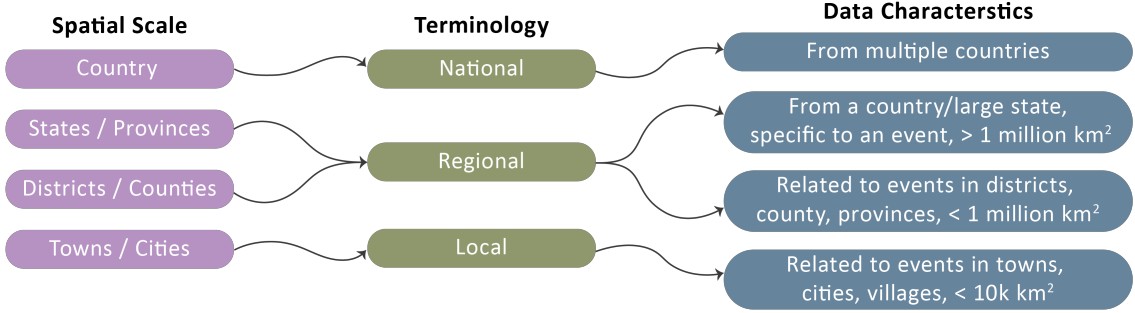

**Figure 4.** Defining study area scope categorically by geographic scale namely, national, regional and local with their respective spatial scale and data characteristics.

methodologies applicable to any disaster and are labeled as 'General Emergency.' Additionally, articles related to social media analytics and social science, not necessarily in the disaster management domain, are categorised as 'Other' in the event type field.

### 3.3.4  Category D - Data Details

In this category, we record the details of data used in the articles within the database under five subcategories, as depicted in Figure 3. Subcategory D1, 'Social Media Used,' indicates whether an article utilises social media data, with options 'Yes' or 'No.' If an article uses social media data, we populate the amount of data, duration under consideration, and the data language in subcategories D2, 'Data Size,' D3, 'Data Duration,' and D4, 'Language,' respectively. Some articles combine social media data with official government records of a disaster event as supporting data, and details of such data are entered in subcategory 230   D5, 'Other Data Used'.

### 3.3.5  Category E - Data Collection Methods

This category extensively examines the data collection and filtering methods employed in each article. We begin by providing an overview of the methodology in subcategory E1, 'Overview.' Subcategory E2, 'Data collection method used,' receives a 'Yes' or 'No' value, indicating whether an article employs a specific data collection methodology. Some articles utilise existing 235   social media corpora rather than developing their own data collection methodology, and this is noted. When an article defines a data collection methodology, a summary is provided in subcategory E3.

Given our focus on relevance filtering of social media data, subcategories E4 and E5, 'Exclusion Criteria Used' and 'Exclusion Criteria Analysis,' indicate whether an article utilises criteria to exclude irrelevant data and whether they analyse their filtering methodology to address the challenge of potentially missing relevant data.



### 3.3.6 Category F - Data Analysis Methods

We document the data analysis methods employed in each article in subcategory F1, 'Data Analysis Method Used,' which may encompass 'NLP,' 'ML,' 'Statistical,' and other methods. The specific algorithms used for each analysis method are detailed in subcategory F2, 'Algorithms Used'.

### 3.3.7 Category G - Evaluation Methods

For each of the algorithms mentioned in Category F2, we document the evaluation methods employed in subcategory G1, 'Evaluation Method Used,' which may include commonly used scoring metrics. The respective average scores for these evaluation methods are recorded in subcategory G2, 'Evaluation Score'.

In the following section, we further analyse the results of the Critical Review Literature Database.

## 4 Results

In this section, we present the results and findings of the social media literature database construction. In the following subsections, we present a detailed analysis across several key dimensions, including early works, publication trends, article classification, data collection methodologies, relevance filtering strategies, and actionable information extraction.

### 4.1 Overview of Literature Database Construction

Our critical review methodology identified a total of 250 articles published in journals, conferences, reports, and books, all of which are listed in the Social Media Literature Database. Figure 5 provides an overview of the total number of papers and the total citations per year from 2010 to September 2023. Approximately 90% of the articles were sourced from Google Scholar, with the remaining 10% obtained from Scopus.

Over the past decade, many authors (Sakaki et al., 2012; Carter et al., 2014; Gunawong and Butakhieo, 2016; Stephenson et al., 2018; Bunney et al., 2018; Brangbour et al., 2019; Brangbour et al., 2020; Podhoranyi, 2021) have conducted experiments in social media data collection and analysis as depicted in figure 5. Initially, while the number of papers was relatively low, there were a significant number of citations. However, in the literature database, we observe a substantial increase in both papers and citations from 2014 to 2018. During the last ten years, a wide range of articles, including journals, conference proceedings, reports, and book chapters, have been published due to the growing use of web data in various phases of the disaster management cycle.

Our critical review encompasses not only journal articles but also conference proceedings, reports, and book series chapters. This choice is driven by the fact that these sources often provide insights into the development of social media data collection, which includes filtering, a core aspect of our review. Figure 6 illustrates the distribution of articles among the categories: 'Journal,' 'Conference,' 'Report,' and 'Book,' with the majority of articles falling under the 'Journal' category. We can also observe that 2018 is the year when journals and conference papers were maximum. Reports and book chapters are comparatively less





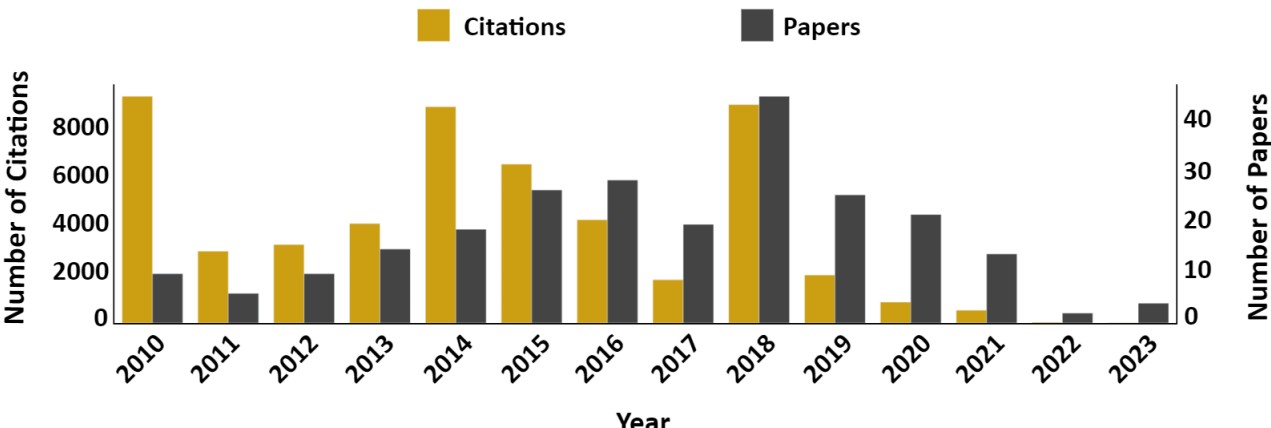

**Figure 5.** The figure shows the number of papers (dark grey, secondary y-axis) and total citations (dark yellow, primary y-axis) per year, in the review duration (2010 to September 2023).

but provide insights into data collection and analysis strategies.

## 4.2 Early Works

This section provides an overview of the early works and discusses the current state of research regarding the use of relevance filtering methods in social media data.

### 4.2.1 Social Media for Disaster Management

From the literature, we have identified four major categories of articles that obtain and utilise User Generated Information (UGI) for disaster management.

  I Articles that obtained UGI based on surveys and questionnaires

  II Articles that discuss why social media data can be considered an effective UGI for managing disasters

III Articles that obtained and utilised social media data for disaster management, particularly Twitter data and,

  IV Articles that use social media data and perform advanced analysis using ML and NLP

In category I, we examine articles that collect UGI through on-site surveys and questionnaires conducted in disaster-affected areas (LÓPEZ-MARRERO, 2010; Aisha et al., 2015; Anson et al., 2017). These responses are subsequently used for enhancing future preparedness, mitigation, and awareness initiatives. A substantial body of research began leveraging public data for dis-

aster management analysis across all phases, primarily through post-disaster surveys and questionnaires (Islam and Walkerden,





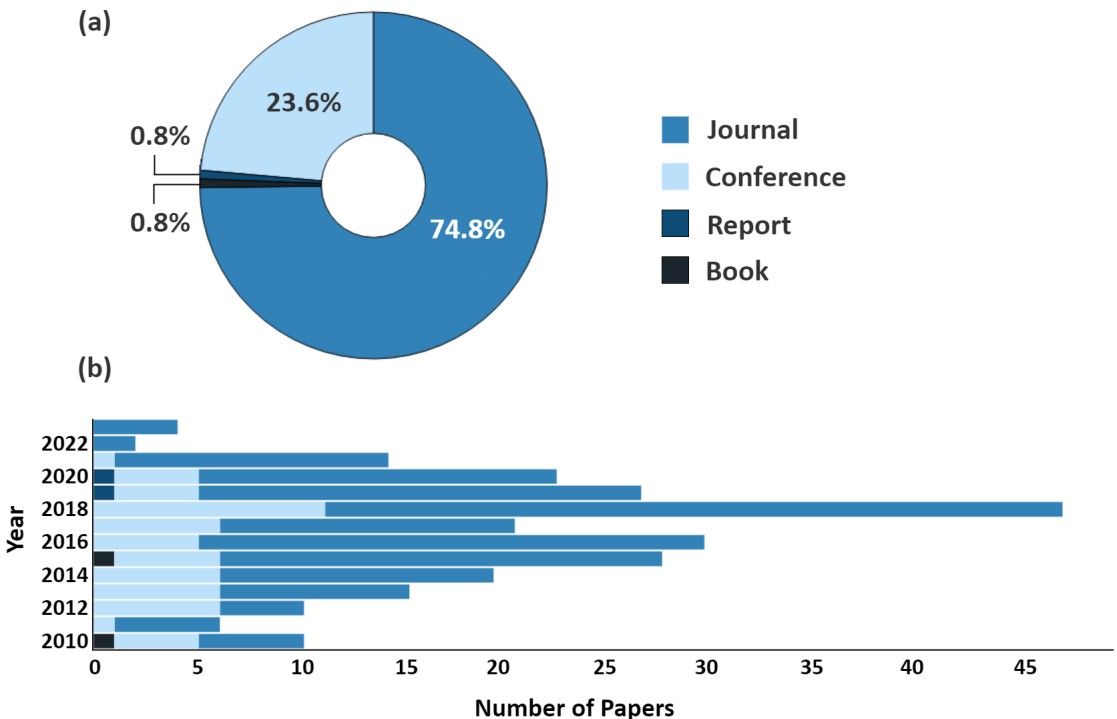

**Figure 6.** The figure shows the number of articles under the categories 'Journal', 'Conference', 'Report', and 'Book'. (a) - The pie chart shows the total percentage distribution of the categories listed in the database; (b) - The horizontal bar chart shows the number of articles under each category year-wise.

2015; Ferris et al., 2016). The overall objective was to generate data that enables government and non-government organisations to collaborate, plan, and execute post-disaster missions and enhance future mitigation efforts. Such data is considered credible when compared to the data extracted from social media platforms as may include false information.

Generally, the questionnaire surveys were administered among local residents and government officials, and school authorities,
to quantitatively assess public awareness of natural disasters and propose recommendations to improve awareness and their impacts. They focused on schools and the local communities to help in future mitigation and preparedness. They also performed damage estimations using the data acquired through surveys and interviews of the affected population and discussed the preventive measures they undertook and the damages caused at their homes.

They also highlighted the potential future directions where such data can be used for effective disaster mitigation and prepared-
ness. The articles under this category indicated that data from actual victims, and government or non-government authorities collected through a questionnaire or a survey is reliable and can be used for future preparedness (Tandoc Jr and Takahashi, 2017; Albris, 2018; Delilah Roque et al., 2020).

In Category II, we explored articles that examined the potential of social media data as a source of UGI for disaster management. During the late 2010s to early 2013s, authors discussed methodologies for collecting social media data and its role in





enhancing post-disaster scenarios (Gao et al., 2011; Abel et al., 2012; Imran et al., 2013b). Notably, the use of platform-specific APIs for data collection was a common practice, alongside the development of open-source platforms that gathered social media data and generated crisis maps, facilitating response and recovery efforts.

These articles also emphasised the active engagement of people on platforms like Twitter and Weibo during disasters and their role in information dissemination. They showcased the utilisation of social media and various free web-based platforms to
enhance awareness of humanitarian assistance and disaster relief. In this category, the focus primarily revolved around considering social media data for disaster management due to its widespread availability, user accessibility, its role in raising hazard awareness and enhancing situational awareness.

In Category III, we reviewed articles that gathered and analysed social media data to enhance disaster management. These articles investigated the utility of social media data through historical disaster case studies, demonstrating its potential for
improving disaster management. Notably, in 2011 and 2012, research began delving into the use of social media data for the analysis of disaster events, with a particular focus on early warning, response, and recovery.

Several authors (Choi and Bae, 2015; Huang and Xiao, 2015; Ogie et al., 2019; Son et al., 2019; Singh et al., 2019; Fan et al., 2020) developed frameworks for collecting and analysing VGI, particularly Twitter data, to improve the reliability of data in crisis management. They explored current trends, identified hotspots, and gathered community-level insights. Their analysis
revealed increased Twitter activity during and after hazard events, with higher post frequencies in proximity to these events. These studies also highlighted the role of emergency communication within the context of social networks, serving as a responsive channel during disasters and fostering a platform for an emergent community of responders and victims.

Since 2013, a substantial body of research (Olteanu et al., 2015; Wang et al., 2016; Han et al., 2020) has concentrated on collecting and analysing social media data to enable rapid decision-making in emergencies. This led to the creation of category IV,
where advanced ML and NLP methods were applied to extract insightful inferences from social media data. During this period, researchers recognized the significance of deep text analysis in identifying pertinent data, considering the diverse languages, informal writing styles, dialects, and the need to preserve contextual nuances in social media posts.

Figure 7 illustrates the evolution of research on the use of social media and web data for disaster management over the years. The articles in consistently affirm the utility of social media data in disaster management while acknowledging its primary
limitation – reliability. In 2010 and early 2011, there was active use of surveys and questionnaires for data collection, which decreased after 2011. Similarly, during this period, articles discussed the potential of social media data for disaster management, with a decline observed after 2011. Categories 3 and 4, focusing on the effective use of social media data and the implementation of ML and NLP methods for deeper analysis, remained active from late 2011 to 2022. It is important to note that some articles may belong to multiple categories, discussing both the relevance of social media and its practical use. Figure
7 provides a detailed representation of this analysis.

### 4.2.2    Previous Works on Social Media Analytics

As part of our research focus on examining the existing relevance filtering methodologies for social media data, we expanded our scope beyond disaster management-related articles to encompass those related to social media analytics. This approach





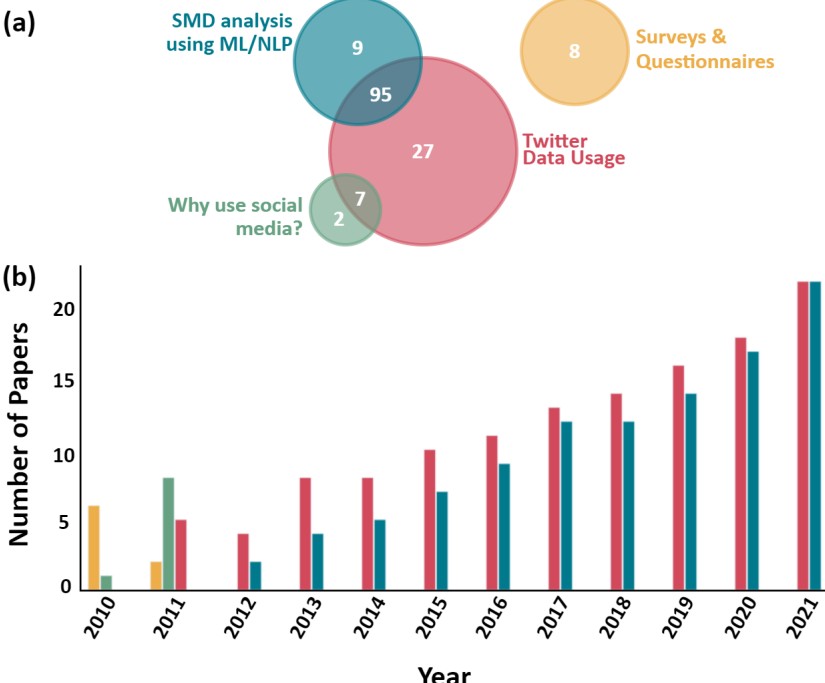

**Figure 7.** The figure summarises the existing usage of social media data for disaster management. (a) - The Venn diagram shows the number of papers in one or more categories; (b) - The grouped bar chart shows the number of papers in each category year-wise.

allows us to cover both the technical aspects of data collection and analysis. Among the 250 articles in the literature database,

15.6% (39 of 250) fall within this category. In this section, we provide a brief overview of a few key articles.

Since 2010, several authors conducted experiments utilising advanced methods for identifying, acquiring, filtering, and analysing relevant data (Gerlitz and Rieder, 2013; Tang et al., 2015; Batrinca and Treleaven, 2015; Steiger et al., 2015; Eilander et al., 2016; Ilieva and McPhearson, 2018; Zhang et al., 2018; Bukar et al., 2022). Approximately 95% of the articles within the literature database relied on Twitter data for their experiments, while the remaining 5% utilised data from other web-based

sources (Facebook, YouTube, Flickr, Weibo) and incorporated public interviews or surveys obtained through manual efforts.

Several authors (Cameron et al.,2012; Black et al., 2012; Oussalah et al.,2013; Schempp et al., 2019; Kejriwal and Gu, 2019; St Denis et al., 2020; de Oliveira and Guelpeli, 2020) developed software frameworks capable of querying the Twitter Streaming API through a series of independent search jobs on an ongoing basis. The extracted data is stored in a well-structured database. These frameworks also describe the query creation process, enabling software to search and collect Twitter data

based on specific criteria, including topic-specific keywords, user-specific posts, location-specific data (defined by bounding boxes), and date-specific parameters. Such architecture proves invaluable in disaster scenarios where precise location-specific data acquisition is crucial (Abel et al., 2012; Muhammad et al., 2018; Yang et al., 2019a; Ghawana et al., 2021).

In the later years, a few authors (Gautam and Yadav, 2014; Wachowicz et al., 2016; Huang et al., 2016; Huang et al., 2018;



Mazoyer et al., 2018; Suzuki, 2019; Brena et al., 2019; Hao and Wang, 2020; Domala et al., 2020; Yuan et al., 2021; Akhter
et al., 2021; Jiang et al., 2022) intensively examined the usage of NLP and AI technologies to analyse the social media data to
extract insightful information that can be applied in various disciplines.

These articles explored the use of public posts that contain opinions and emotions to facilitate solution development. They
conducted feature extraction by examining adjectives in tweets, which reflected the sentiment of individuals as positive, nega-
tive, or neutral. The articles employed probabilistic classifiers like Naive Bayes and maximum entropy to determine sentiment.
Sentiment analysis holds great relevance in disaster management scenarios as it contributes to community analysis and feed-
back during or after a disaster (Mandel et al., 2012; Neppalli et al., 2017; Ragini et al., 2018; Wu and Cui, 2018; Reynard and
Shirgaokar, 2019; Pourebrahim et al., 2019; Yabe and Ukkusuri, 2019; Karimiziarani and Moradkhani, 2023).

Similarly, some authors investigated the behaviours of social media users engaged in the consumption and dissemination of
news items (Lachlan et al., 2010; Houston et al., 2012; Liu and Stevenson, 2013; Kaewkitipong et al., 2016; Valenzuela et al.,
2017; Kim et al., 2018; Verma et al., 2019; Yeo et al., 2022). They provided detailed insights into the distribution and statistics
of Twitter users who shared or retweeted topic-relevant news articles. Behavioural analysis of the public on social media holds
significance in crisis management, as it enables an examination of information dissemination patterns (Mendoza et al., 2010;
Kim, 2014; Chae et al., 2014; Spence et al., 2015; Hara, 2015; Kibanov et al., 2017; Jitkajornwanich et al., 2018).

While investigating public behaviour, several researchers deemed the language usage of social media users as relevant (Lee
et al., 2011; Abel et al., 2012; Reuter and Schröter, 2015; Carley et al., 2016a; Xu et al., 2019b). They argued that enhancing
methods for the analysis of non-English social media data is essential to improve global community interaction. Various en-
sembles of ML models and NLP algorithms have been utilised to analyse non-English content. Each of these articles discussed
how every language varies in writing style, grammar, and local usage. In disaster scenarios, the inclusion of posts written in
local languages aids in analysing community-level needs.

We observed that, over the past decade, researchers have increasingly advocated the use of NLP and AI technologies for social
media data acquisition and analysis across various disciplines. However, a recurring challenge in every article is the issue of
data reliability, which has prompted the inclusion of supplementary data sources such as government records or news reports
to substantiate their findings.

### 4.3   Actionable Information (AI) Analysis

To address our research question "Does social media provide actionable information (AI) in disaster scenarios?", we analyse
the articles listed in the Social Media Literature Database under the theme of "Disaster Management". By studying various
articles (Palen et al., 2010; Sakaki et al., 2010; Zhou et al., 2013; Jongman et al., 2015; Musaev et al., 2018; Phengsuwan et al.,
2019; Guntha et al., 2020b; Gopal et al., 2020; Guntha et al., 2020a; Gopal et al., 2022; Aswathy et al., 2022) and based on
our experience, we have defined nine generic actionable information categories which will be assigned to each article under
the "Disaster management" theme listed in the literature database.

An article can fall into multiple AI categories, and Table 1 outlines these nine categories. These classifications center around
data collection methods, geolocation identification, relevance filtering strategies, community and stakeholder collaborations,




and software development. Table 1 displays the various categories with their respective descriptions, detailing the methods and applications considered within each AI category in this study. Additionally, we include references for articles under each AI

category that have garnered higher citations compared to others in the same category.

Table 1: Actionable Information (AI) analysis - table describes the 9 AI categories. Each article listed in the Social Media Literature Database belonging to the "Disaster Management" theme gets grouped under one or more AI categories. The 'References' column shows the articles with high citations under an AI category for reference.

| AI | Category | Description | References |
|---|---|---|---|
| 1 | Disaster Data Collection | APIs or other programming used for data collection, temporal or distribution analysis performed, explores communication of the public over the course of the disaster. | Middleton et al. (2013), Radianti et al. (2016), Granell and Ostermann (2016) |
| 2 | Geolocation Detection and Analysis | Methods of geolocation identification or geocoding from tweet content or user details or from geotagged data, performs spatial analysis of their respective research. | Kryvasheyeu et al. (2016), Cenni et al. (2017), Kankanamge et al. (2020b) |
| 3 | Relevance Filtering | Uses appropriate keywords for collecting data, irrelevant data exclusion (advertisements, social media posts discussing older disaster events that has no relevance in the current events, exclude topics with word similarity, but context dissimilarity), identifies fake data (rumoured disaster events, retweets of rumoured posts) | Campan et al. (2018), Abedin and Babar (2018) |





Table 1: Actionable Information (AI) analysis - table describes the 9 AI categories. Each article listed in the Social Media Literature Database belonging to the "Disaster Management" theme gets grouped under one or more AI categories. The 'References' column shows the articles with high citations under an AI category for reference.

| AI | Category | Description | References |
|---|---|---|---|
| 4 | Community Collaborations | Methods of SMD usage in improving community awareness, disseminating or identifying information such as methods of preparation - rescue camps, sources of food and medication, financial needs, financial claims, emergency information, examining community interactions over social media, tracking people's emotions pre/during/post a disaster occurrence. | Plachouras et al. (2013), Olteanu et al. (2014), Verma et al. (2019) |
| 5 | Disaster trends and hotspot identification | Analyses previous disaster events of various locations from social media data, Studies current landscape of a location based on geography, demography or development/urbanisation, predicts the probability of the disaster to occur again in future, Identifying disaster events or performs disaster relevant topic modelling or event classifications. | De Albuquerque et al. (2015), Fohringer et al. (2015) |



Table 1: Actionable Information (AI) analysis - table describes the 9 AI categories. Each article listed in the Social Media Literature Database belonging to the "Disaster Management" theme gets grouped under one or more AI categories. The 'References' column shows the articles with high citations under an AI category for reference.

| AI | Category | Description | References |
|----|----------|-------------|-----------|
| 6 | Stakeholder Identification and Collaboration | Identifies community stakeholders, government stakeholders, non-government organisation stakeholders, inter-governmental organisations and volunteers; establishes a strategy for stakeholder network creation; defining methods of collaborative crisis management within stakeholders; generates network graphs and analyses communications between stakeholders/public. | Imran et al. (2013a), Ragini et al. (2018) |
| 7 | Open-source software development | Software tool/dashboards/websites/apps, Tool that provides real time or near real time alerts, warnings | Lachlan et al. (2010), Ashktorab et al. (2014), Jitkajornwanich et al. (2018) |
| 8 | Resource Identification | Extracts the needs of the public from social media posts, establishes a strategy to be employed by the organisations to allocate resources based on the needs - includes rescue missions, food/water/ medication/clothing requirements, performs damage/risk assessment. | Lee et al. (2011) |
| 9 | Community Response Analysis | Response/feedback messages from the public post a requirement fulfilment from concerned authorities, behavioural analysis of public, interviews or surveys taken. | Chen et al. (2014), Reuter and Schröter (2015), Huang et al. (2016) |





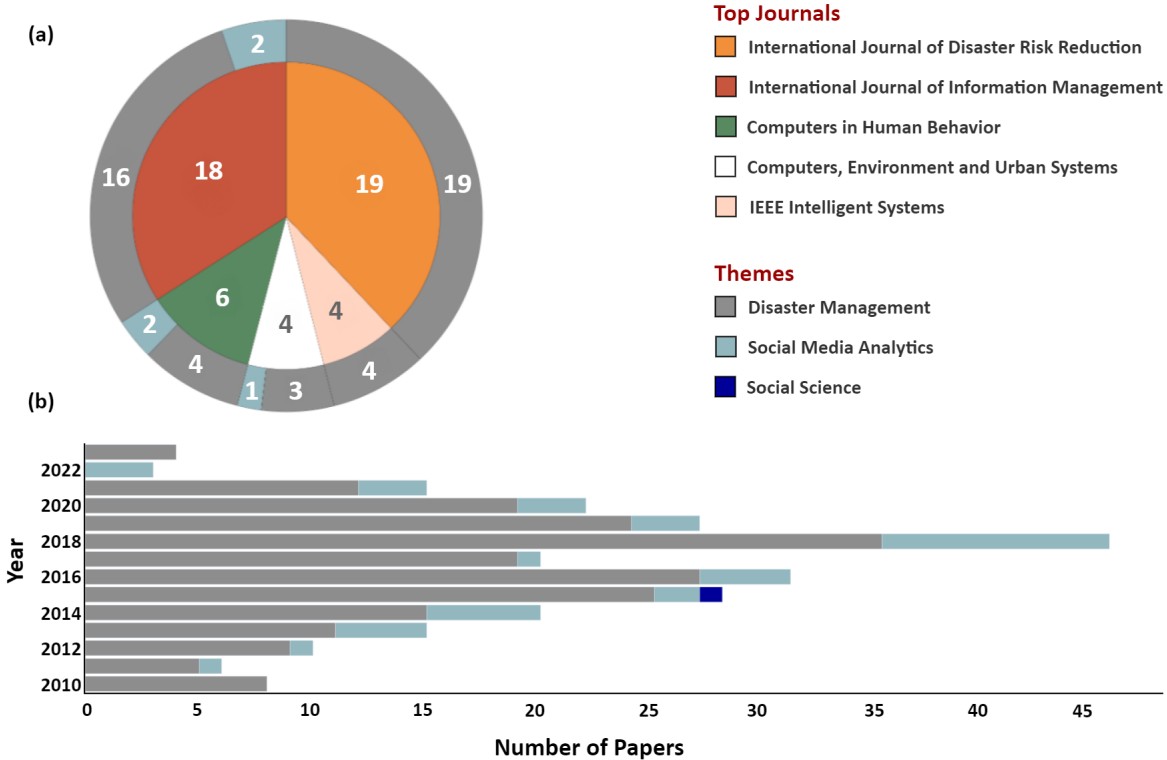

**Figure 8.** The figure shows the classification of articles based on themes 'Disaster Management', 'Social Media Analytics', and 'Social Science' and based on top journals. (a) - The sunburst chart shows the number of articles under the top 5 journals and is further classified under each theme category. The inner circle shows the former and the outer circle shows the latter. (b) - The horizontal bar chart shows the number of articles under each theme category year-wise.

## 4.4  Journal Distribution and Theme Analysis

Among the 250 articles included in the literature database, 184 were published in journals. Figure 8 provides insight into the top 5 journals, which account for 2.98% of the journals included in the database, and the number of articles published in each of these journals. We have further classified these articles under the themes 'Disaster Management,' 'Social Media Analytics,'

and 'Social Science.'

Figure 8 presents the statistics for this classification as well, where the 'Disaster Management' theme is maximum. The figure also shows the number of articles on a particular theme falling under a top journal. We can observe that the 'Social Science' theme is the least in comparison to the other categories. However, an article belonging to the 'Social Science' theme aids in understanding how demographic study can be done using social media data. Similarly, the articles belonging to the 'Social

Media Analytics' show various methods of social media data collection from a development perspective.





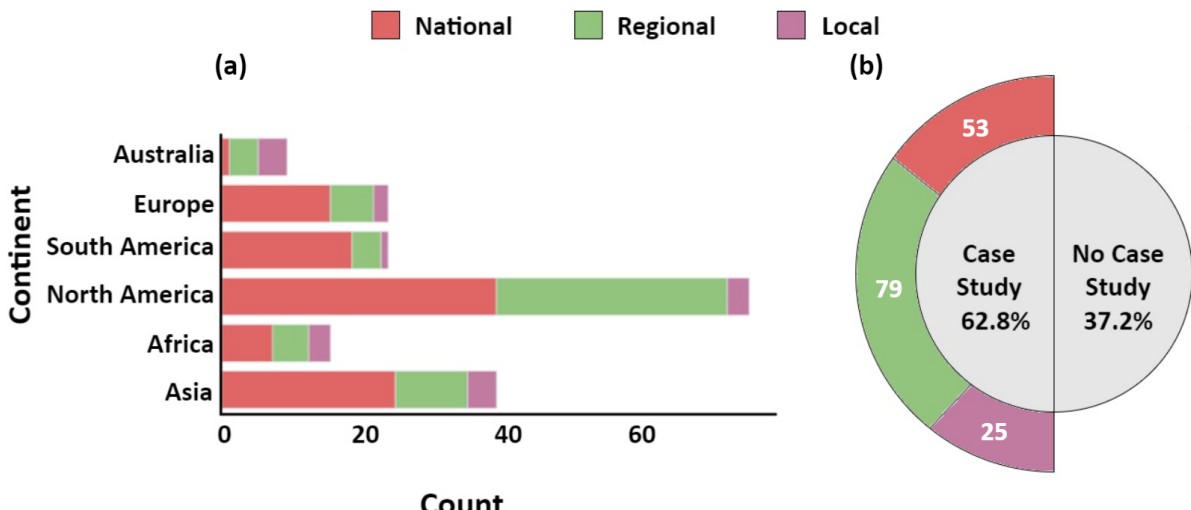

**Figure 9.** The figure shows the articles that use a case study or not. (a) - The bar graph shows the number of articles that use a case study area categorised continent-wise. (b) - In the sunburst chart the inner circle represents the percentage of papers that use a case study or not; the outer circle represents the number of articles that use a case study categorised under each scope.

## 4.5 Case Studies and Geographic Scope

Over 60% of the articles (Jung et al., 2015; Chen et al., 2016; Bala et al., 2017; Kurkcu et al., 2017; Bala et al., 2017; Liu et al., 2018; Brangbour et al., 2020; Rahmadan et al., 2020; Liu et al., 2020) categorised under the 'Disaster Management' theme employed case studies to evaluate their methodologies.

Figure 9 provides statistics on the use of case studies and their geographical scope. Notably, approximately 50% of the articles (Yuan and Liu, 2018; Abirami et al., 2019; Yang et al., 2019b; Kankanamge et al., 2020a) utilised regional case studies, which was the most prevalent among the different geographical scopes. It is worth mentioning that North America, particularly events such as Hurricane Sandy (2012), Hurricane Matthew (2016), and the Red River Valley Flood (2009), was the most frequently used region in these case studies (Ferris et al., 2016; Martín et al., 2017). We can also observe from the figure that around 37%

of the articles do not use a case study to validate their respective methodologies.

## 4.6 Disaster Events

Articles categorised under the 'Disaster Management' theme were further classified under the 'Event' section, indicating the specific disaster events studied by the respective authors. Figure 10 shows the statistics of the 'Event' category of the literature database. Our examination of these case studies revealed that 'Flood' was the most frequently studied disaster event, followed

by 'Hurricane' event (Middleton et al., 2013; Freberg et al., 2013; Gupta et al., 2013; Guan and Chen, 2014; Xiao et al., 2015; Yoo et al., 2016; Jamali et al., 2019; Wang et al., 2019). We can also observe that the 'Earthquake' event was studied every year of the review period. The least studied events were 'Storm', 'Volcanoes' and 'Cyclone'.





**Figure 10.** The figure shows the analysis of the articles categorised under the 'Disaster Management' theme – The large bar chart represents the number of articles under various disasters used as a case study. The smaller bar charts show the year-wise statistics of each disaster event.





## 4.7 Data Sources and Collection Methods

Among the articles listed in the literature database, excluding the review articles, approximately 72% (182 out of 250) utilised
social media data from various platforms as their input data (Gautam and Yadav, 2014; Uchida et al., 2016; Branz and Brock-
mann, 2018; Alampay et al., 2018). Within this category, 70% of the articles developed their own methodologies for collecting
social media data tailored to their specific needs (Driscoll and Walker, 2014; Gaspar et al., 2016; Mac Kim et al., 2016; Healy
et al., 2017; Campan et al., 2018). They frequently employed APIs, such as the Twitter Streaming API and REST API. The
remaining 2% of the articles utilised social media data available as online resources from various portals (Ai et al., 2016;
Madichetty and Sridevi, 2021).

Out of the 250 articles, excluding the review papers, nearly 13% (34 articles) sourced their data from government authority
portals (Ofli et al., 2016; Williams et al., 2018). Frequently accessed portals included FEMA (Federal Emergency Manage-
ment Agency, USA) and USGS (United States Geological Survey), which offered valuable disaster-related social information,
satellite image data, and historical event damage data. Additionally, approximately 4% (11 articles) of the total used manually
collected interview or survey data (Adam et al., 2012; Aisha et al., 2015; Le Coz et al., 2016; Lin et al., 2018; Lu and Yuan,
2021).

## 4.8 Data Relevance Filtering

The identification of relevant data presents a significant challenge in social media data collection. The majority of articles,
around 70%, employed NLP-based methods, particularly text analysis, to address this challenge (Starbird et al., 2010; Terpstra
et al., 2012; Panagiotopoulos et al., 2016; Laylavi et al., 2017; Lin et al., 2018). These methods involved the use of inclusion
keywords specific to their topics of interest during data collection. While this approach aids in identifying topic-relevant data,
it may also introduce a considerable amount of noise.

The use of exclusionary criteria proved valuable in noise reduction, with approximately 12% of the articles adopting this
approach (Joseph et al., 2014; Radianti et al., 2016; McCormick et al., 2017). These articles utilised NLP and ML-based
solutions to exclude irrelevant data. Exclusionary criteria are often constructed based on assumptions, emphasising the need
for rigorous evaluation before concluding. However, only a small percentage, approximately 2% of the articles, conducted such
evaluations before proceeding with the data analysis (Spinsanti and Ostermann, 2013; Herfort et al., 2014; Li et al., 2018a;
Ahmad et al., 2019).

In Figure 11, we present a summary of the relevance filtering analysis from the articles in the literature database. We can
observe that only 14% of the articles used exclusionary criteria to perform relevance filtering. Notably, the majority of the
articles employed NLP methods to perform filtering in comparison to ML methods. This analysis allowed us to answer our
research question (Q1), demonstrating that performing relevance filtering is vital for improving data quality and application
effectiveness. We recommend a thorough study of input data and the implementation of NLP or ML methods for effective
relevance filtering strategies.





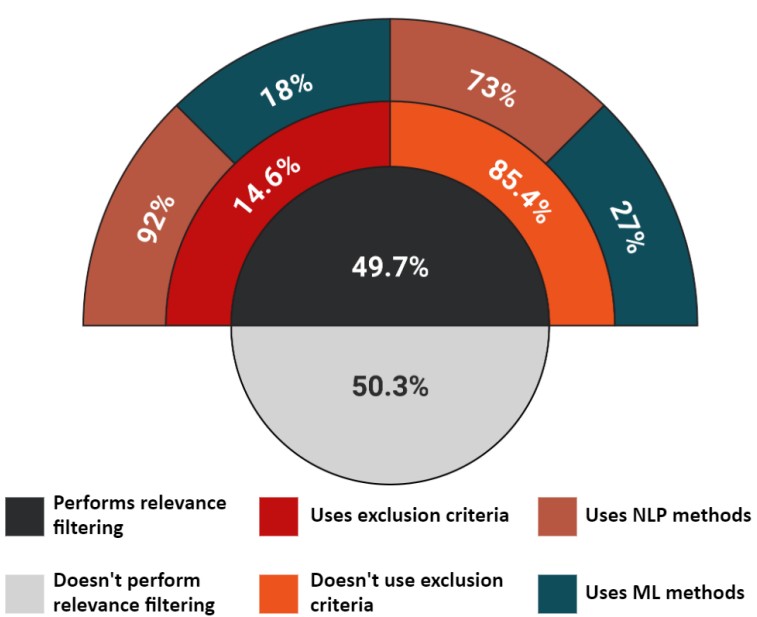

**Figure 11.** Result of research question Q1 - The chart shows the percentage of articles under each legend category summarising the data filtering methods employed in the articles listed in the literature database.

## 4.9 Data Analysis Methodologies


The methodologies employed in the articles within the literature database encompass a range of techniques in the fields of NLP and ML. These methodologies include text analysis, Named Entity Recognition (NER), Bag-of-Words (BoW), Part-of-speech Tagging (PoS), and various feature extraction methods. Data analysis is carried out using both supervised and unsupervised ML models, employing algorithms such as Logistic Regression (LR), Support Vector Machines (SVM), Naive Bayes (NB), K

Nearest Neighbors (KNN), Convolutional Neural Networks (CNN), Decision Trees (DT), Random Forest (RF), Latent Dirichlet Allocation (LDA), and more (Plachouras et al., 2013; De Albuquerque et al., 2015; Zhang et al., 2019b).

Some articles employ statistical techniques, including correlation analysis (e.g., Pearson's and Kendall's), distribution analysis (e.g., Poisson and Binomial), and Generalised Additive Models (GAM) (LÓPEZ-MARRERO, 2010; Liu and Lee, 2010; Lu and Yang, 2011; Yin et al., 2012; Westerman et al., 2014). Others explore methodologies that establish relationships among

stakeholders in disaster scenarios and conduct network analyses to enhance decision-making in the wake of disasters (Kogan et al., 2015; Wang et al., 2016; Htein et al., 2018; Kim and Hastak, 2018; Rajput et al., 2020; Wang et al., 2021). Figure 12 provides statistics on the technologies featured in the reviewed articles.

From figure 12 we can observe that NLP methods were employed the most where text analysis was in the majority. Analysing the text of the social media post helps in identifying topic-relevant keywords, event location, duration of the event, and senti-

ment of the user. ML methods were also used for analysis and the SVM algorithm was found frequently used by the investiga-



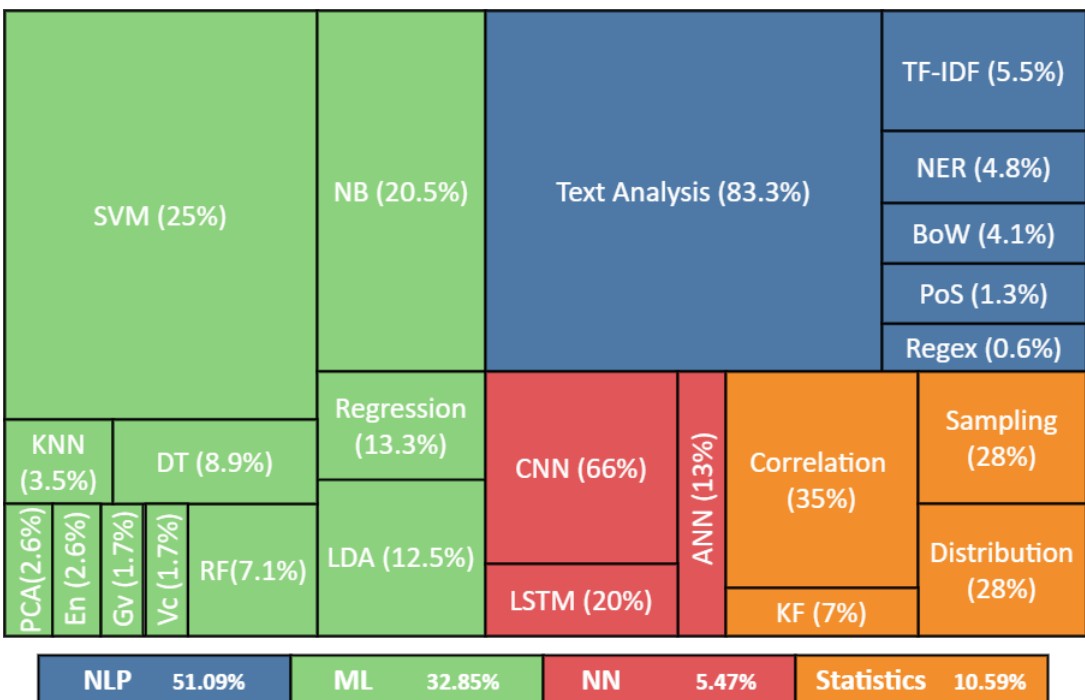

**Figure 12.** The treemap shows the algorithms used in the methodologies defined in the articles listed in the database. The percentage represents the percent of articles that use a particular algorithm (for abbreviations, refer to appendix A).

tors. However, neural network algorithms were not used much in the literature duration.

Roughly 65% of the articles in the literature database conduct performance evaluations using a range of methods. Articles employing ML algorithms often rely on scoring metrics like accuracy, precision, recall, and F-score (Imran et al., 2013a; Olteanu et al., 2014; Wang et al., 2016; Nguyen et al., 2017). Those exploring sentiment analysis in SMD typically utilise polarity scores
for evaluation (Bala et al., 2017; Yuan et al., 2021). Some articles also employ statistical tests, such as ANOVA, chi-square, correlation values, and invariance tests to validate their methodologies (Steelman et al., 2015; Reuter and Spielhofer, 2017). Additionally, a few authors opt for manual evaluations (Stephenson et al., 2018; Liu et al., 2020).

### 4.10 Actionable Information

The articles categorised under the 'Disaster Management' theme were categorised further based on the Actionable Information
(see table 1) classes to address our research question Q2. Figure 13 shows the number of articles assigned to each AI category year-wise and on an overall basis.

From figure 13 we can observe that AI-1 (95 of 211) (Howe et al., 2011; Chae et al., 2012; Fohringer et al., 2015), AI-2 (91 of 211) (McClendon and Robinson, 2013; Abedin and Babar, 2018; Boas et al., 2020) and AI-3 (121 of 211) (Castillo et al., 2013; Neppalli et al., 2017; Madichetty, 2020) was the most prevalent where articles focused on the development and testing of social





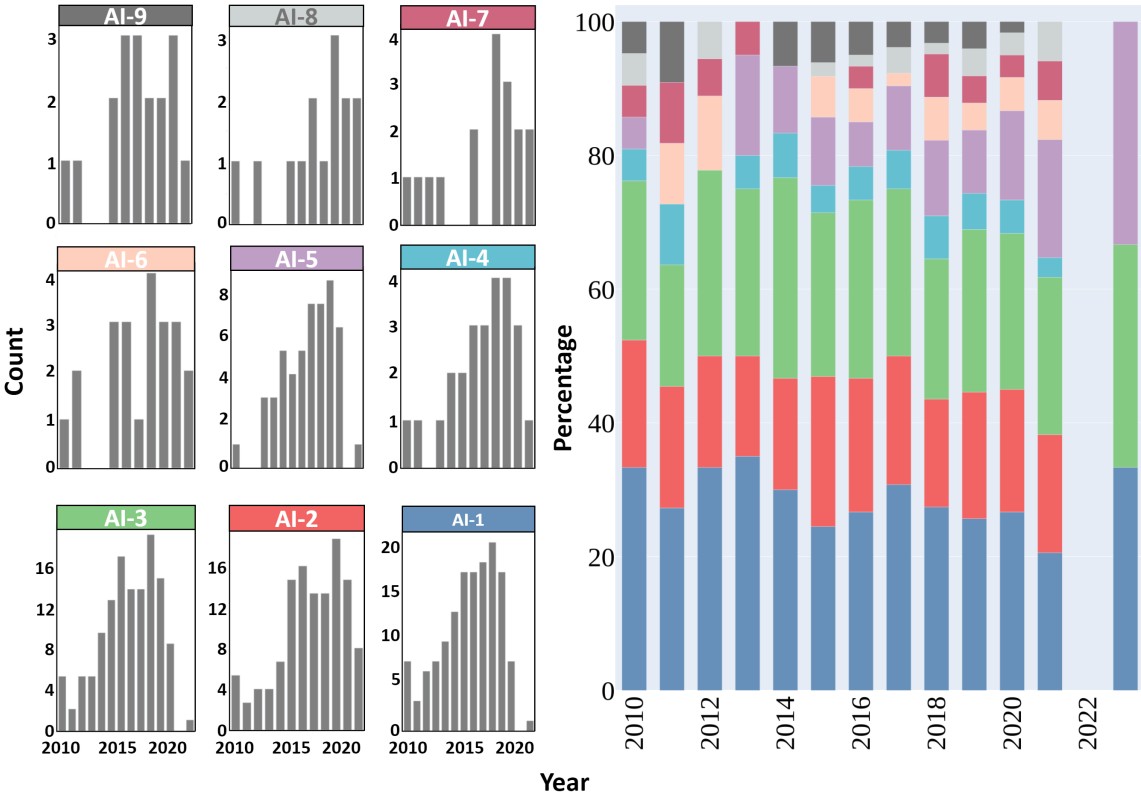

**Figure 13.** The figure shows the analysis of the disaster management-related articles categorised under the 'Disaster Management' theme - The stacked bar chart summarises the percentage of AI classes yearwise and the smaller bar charts show individual summaries of each AI class yearwise.

media data collection methodologies and geolocation identification methodologies and conducting spatial analyses. Notably, 57% of the articles were classified under AI-3, emphasising the significance of relevance filtering in disaster scenarios, and investigating methods to enhance data quality and reduce noise.

Around 11% (25 of 211) fall under AI-4 which studied how SMD can be utilised for community collaborations (Yang et al., 2019b; Yuan et al., 2021). Approximately 23% (50 of 211) and 10% (22 of 211) fell under AI-5 (Podhoranyi, 2021; Karim-

iziarani and Moradkhani, 2023) and AI-6 (Htein et al., 2018; Delilah Roque et al., 2020) respectively, which focused on the identification of disaster trends, disaster hotspots, and stakeholder collaborations. Around 8% articles (17 of 211) developed open-source software and were categorised under AI-7 (Yuan and Liu, 2018; Podhoranyi, 2021).

AI-8 (14 of 211), involving resource identification methodologies, received the least attention (LÓPEZ-MARRERO, 2010; Houston et al., 2012; Delilah Roque et al., 2020). Articles categorised under AI-9 (18 of 211) focused on studying the commu-

nity response (Jitkajornwanich et al., 2018; Ahmad et al., 2019).

The analysis indicates that current methods, such as NLP and ML, effectively aid in filtering social media data for relevance,



reducing noise, and excluding irrelevant content. However, challenges related to data reliability, including rumours and false information, persist. Many data collection methods employ inclusion keywords for relevance, which can introduce noise. The use of exclusion criteria proves valuable in enhancing efficiency by eliminating specific data.

Each article categorised under the 'Disaster Management' theme fulfilled at least one actionable information category. Several articles (Cervone et al., 2016; Schempp et al., 2019; Podhoranyi, 2021) met more than five actionable information categories, demonstrating their valuable contributions to efficient disaster management.

## 5    Discussion

In this section, we analyse and discuss the different categories and the corresponding information within the Social Media
Literature Database (Gopal et al. (2023)). We organise this section into subsections to address the various categories within the Social Media Literature Database. We discuss the data collection methods used in the articles (section 5.1), major disaster events used as case studies in the articles (section 5.2), social media data reliability and external data usage in the article methodologies (section 5.3), algorithms used in the article methodologies (Section 5.4), actionable information in the articles (section 5.5), methodological biases (section 5.7), best practices of social media usage (section 5.8) and the practical applica-
tions of the Social Media Literature Database (section 5.9). Additionally, in section 5.6, we delve into a methodology based on our previous work for effectively collecting social media data through the use of exclusion criteria and other NLP techniques.

### 5.1    Data Collection Methods Used in Database Articles

The majority of the 250 articles in our Social Media Literature Database (Gopal et al. (2023)), roughly 70%, devised their methods for collecting social media data, primarily relying on topic-relevant inclusion keywords to sift through the extensive
web resources (Wendt et al., 2016; Henry, 2021). The investigators used the approach of in-depth studying of the topic of interest to identify related keywords which helps to identify relevant content. In such an approach, the major challenge is noise filtering.

For instance, one of the most frequently studied disaster events in the database was Hurricane Sandy, which struck the USA in 2012 (Neppalli et al., 2017; Wang et al., 2019). Many of the articles that focused on Hurricane Sandy as their case study
employed a set of inclusion keywords, including terms like 'Sandy,' 'Hurricane,' 'rainstorm,' 'New York,' 'Caribbean,' 'USA,' and '2012,' along with relevant hashtags. While these keyword sets were effective in gathering data related to Hurricane Sandy from social media platforms, they also resulted in significant noise, such as posts mentioning 'hurricane of emotions' (Spence et al., 2015; Kogan et al., 2015).

Some authors experimented with data collection using exclusion keyword sets, effectively reducing noise. For instance, Mc-
Cormick et al. (2017) collected Twitter data to analyse demographics for social science research, and they used an exclusion keyword set related to 'TV shows' to filter out unrelated data. Similarly, Aswathy et al. (2022) collected tweets and news reports related to disasters and employed an exclusion keyword set that included terms like 'Songs,' 'Election,' and 'Victory' to eliminate irrelevant data, such as mentions of 'Landslide Victory' (part of previous work, discussed in detail in section 5.6).





This approach demonstrates the effectiveness of exclusion keywords in improving data collection efficiency.

Various authors have employed machine learning methods for noise filtering. They utilized supervised classification algorithms to determine the relevance of social media posts to a specific topic. However, a significant amount of data is needed for training, along with experts to prepare the data, making the process time-consuming and requiring ongoing maintenance (Chen et al., 2014; Ghani et al., 2019; Fan et al., 2021).

Our analysis has highlighted the importance of avoiding noise in social media data collection through the use of exclusionary
criteria. However, it is crucial to evaluate these methods to ensure that relevant data is not missed. We recommend that researchers thoroughly study the topic and related data before constructing their methodologies and consider including exclusion criteria to filter out noise at the initial stage. Excluding irrelevant data can significantly impact both time and space complexities and enhance the overall efficiency of the model.

## 5.2 Major Disaster Events in Database Articles

Around 74% of the 250 articles categorised under the 'Disaster Management' theme in the Social Media Literature Database used various disaster events around the globe as their case studies to evaluate their respective methodologies. Figure 14 shows some of the significant historical disaster events in the database. These articles devised methods to extract location information from social media data for conducting case studies. Social media platforms often offer API support that allows for the retrieval of location-specific data using bounding boxes (Purohit et al., 2014; Neubaum et al., 2014). However, challenges arise when
social media users input non-existent place names in their content or profiles, leading to data collection inaccuracies. To mitigate this, some authors exclusively gathered geotagged social media posts, which are considered reliable as they represent the user's present location based on their device settings (Steelman et al., 2015; Resch et al., 2018; Leon et al., 2018; Wang et al., 2021).

Figure 14 illustrates that hurricane events were frequently chosen as case studies by multiple authors. In the 'Disaster Manage-
ment' theme, several articles (Gupta et al., 2013; Neubaum et al., 2014; Steelman et al., 2015; Olteanu et al., 2015; Mukkamala and Beck, 2016; Jamali et al., 2019) utilised hurricane Sandy, a 2012 disaster in the US, as a case study due to the extensive social media activity related to this event. However, assessing the reliability of the content in such scenarios presented challenges, leading to research endeavours aimed at identifying and analysing fake data circulated on Twitter and examining their impact on the public (Pourebrahim et al., 2019; Wang et al., 2019).

Smith et al. (2017) presented a significant article that introduced a real-time monitoring framework for identifying flooded areas using social media data. They evaluated their framework by applying it to the 2012 Tyne and Wear flood as a case study. Another notable event was the devastating Nepal earthquake of 2015. In a pivotal study, Radianti et al. (2016) devised a data collection methodology, that enables the collection, filtering, and categorization of tweets in multiple languages. This methodology aimed to identify disaster response issues, including resource requirements and damages, associated with the
Nepal earthquake.

The 2018 Woolsey fire, which ravaged areas in Los Angeles and Ventura counties, was a significant fire event documented in the database. St Denis et al. (2020) conducted a study on this event, focusing on the behaviour and content of local residents in



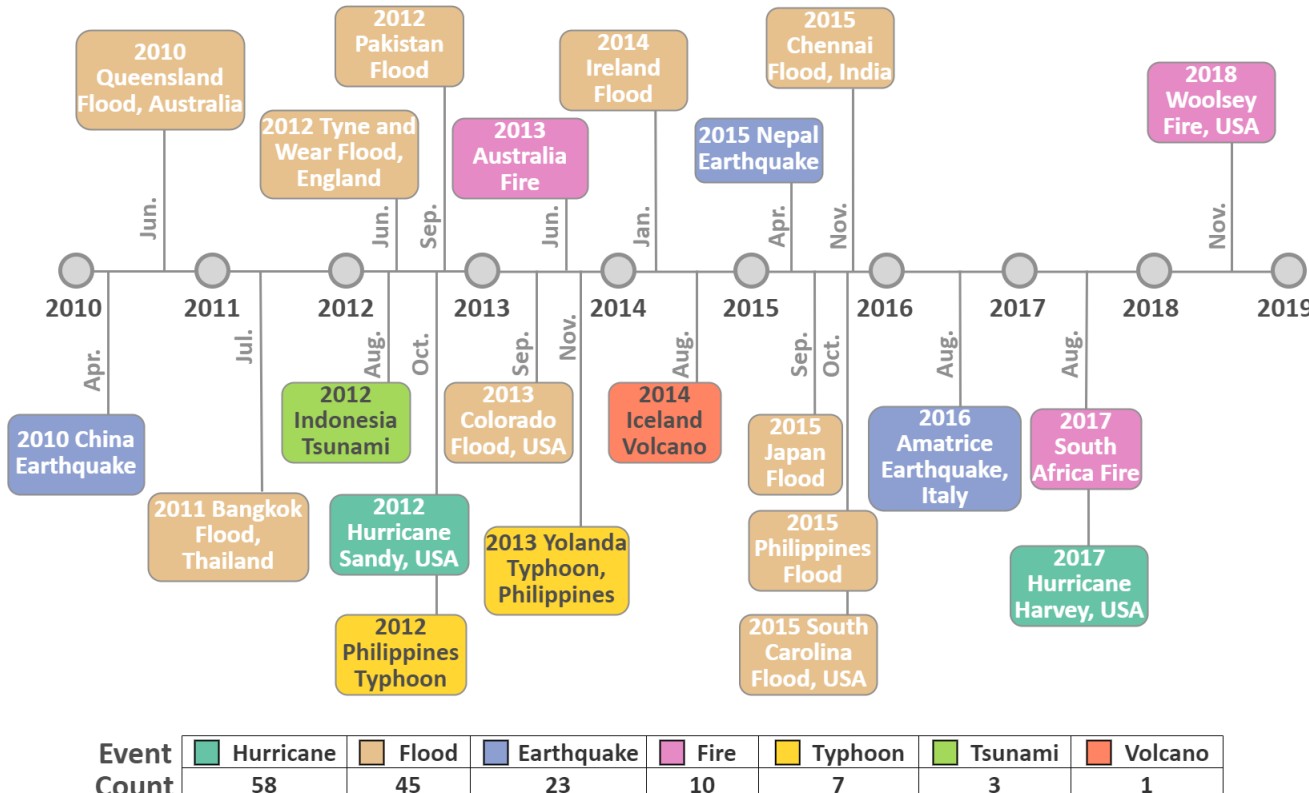

**Figure 14.** The timeline shows some of the significant disaster events that occurred from 2010 to 2019. The legend shows the number of articles that used a particular disaster as their case study.

the affected area. Their research aimed to capture valuable information that could otherwise go unnoticed.

The events depicted in Figure 14 have had significant impacts on the affected populations. To gain a better understanding

of the scale of these disasters, we collected and analysed data from the EM-DAT database to assess the number of affected individuals. Figure 15 presents the statistical findings, revealing that the 2012 Hurricane Sandy in the USA and the 2010 China earthquake each affected more than 20 lakh people.

As demonstrated in Figure 9 (refer to Section 4.5), our analysis of continent-based case studies revealed that North America was the most frequently utilised region by the authors in the Social Media Literature Database. Furthermore, it was evident

that major disaster events generated more data and garnered increased attention on social media platforms. In light of these observations, we recommend that researchers explore methodologies using local disaster events, enabling a more comprehensive examination of data to identify relevant content, manage unfamiliar place names, and enhance disaster management efforts effectively.





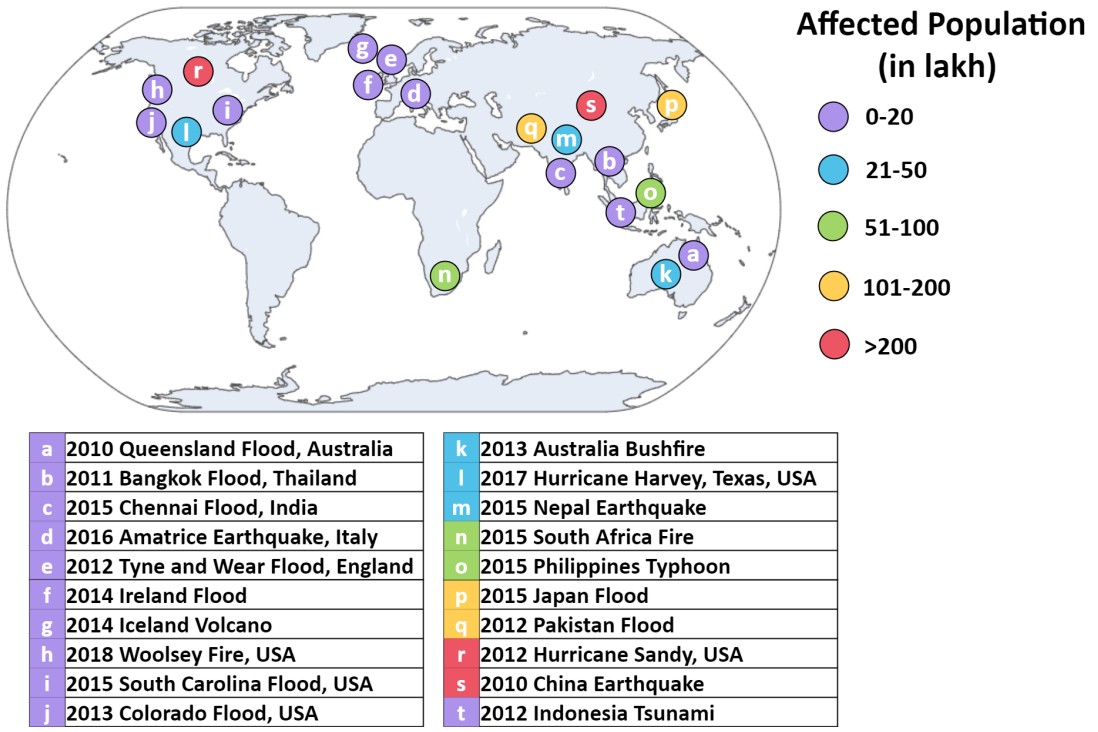

| | | | | |
|---|---|---|---|---|
| a | 2010 Queensland Flood, Australia | k | 2013 Australia Bushfire |
| b | 2011 Bangkok Flood, Thailand | l | 2017 Hurricane Harvey, Texas, USA |
| c | 2015 Chennai Flood, India | m | 2015 Nepal Earthquake |
| d | 2016 Amatrice Earthquake, Italy | n | 2015 South Africa Fire |
| e | 2012 Tyne and Wear Flood, England | o | 2015 Philippines Typhoon |
| f | 2014 Ireland Flood | p | 2015 Japan Flood |
| g | 2014 Iceland Volcano | q | 2012 Pakistan Flood |
| h | 2018 Woolsey Fire, USA | r | 2012 Hurricane Sandy, USA |
| i | 2015 South Carolina Flood, USA | s | 2010 China Earthquake |
| j | 2013 Colorado Flood, USA | t | 2012 Indonesia Tsunami |

**Figure 15.** The figure shows the statistics of the affected population of various historical disaster events. The colour legend represents the affected population and the alphabet legend shows the details of the disaster event plotted on the map.

## 5.3 Social media reliability and usage of external data in Database Articles

The issue of reliability is a significant concern when using social media data for disaster management (Mazoyer et al., 2018; Liu et al., 2020). To address this concern, many authors incorporated external data sources alongside social media data, aiming to enhance the efficiency and reliability of their methodologies (Chatfield and Brajawidagda, 2013; Joseph et al., 2014; Musaev et al., 2018).

External data sources encompass information available on government portals, such as damage-related data, satellite images of
affected areas, disaster event statistics, precipitation data, GIS data, news reports about disaster events, and interview or survey data, among others. Out of the 250 articles in the Social Media Literature Database, 26% (65 articles) integrated external data to enhance their methodologies. Notably, the majority of external data utilisation was observed in articles focusing on events in the USA, where FEMA and USGS data were frequently employed (Hodas et al., 2015; Liu et al., 2018; Musaev et al., 2018).

External data played a vital role in not only collecting additional trustworthy information but also in validating the methodolo-
gies by enabling comparisons of results (Earle et al., 2011; Li et al., 2018b). This approach helped investigators demonstrate the effectiveness of their methodologies in disaster management. Furthermore, it allowed them to identify and refine keyword sets for data collection and uncover place names and other location-specific information that may not be readily available in social





media posts (Dashti et al., 2014; Kryvasheyeu et al., 2016). As a recommendation, we suggest that researchers continue to leverage trustworthy external data sources in conjunction with social media data to obtain more authentic and reliable insights

for disaster management.

## 5.4 Algorithms used by Database Articles

In section 4.9, we explored the algorithms utilised in the methodologies of the articles within the database, as illustrated in Figure 12. These algorithms were categorised into 'NLP,' 'ML,' 'Statistical,' and 'Neural Networks.' Notably, NLP algorithms were the most frequently employed and were consistently utilised across the review period. NLP methods played a crucial role

in content analysis during data collection and filtering (Gupta et al., 2013; Madichetty, 2020).

Statistical methods were primarily used for distribution and correlation analysis, providing valuable insights into the data (Htein et al., 2018; Wang et al., 2019). Starting in 2013, ML methods gained popularity among researchers and were extensively applied to classification, clustering, and filtering tasks. Of the various classification algorithms, SVM was commonly employed in the literature and was found effective. Additionally, neural network algorithms began to gain traction around the same time

and exhibited impressive performance in various applications (Neppalli et al., 2017; Reynard and Shirgaokar, 2019).

Figure 16 presents statistics on the methodology categories, offering an overview of overall percentages and a year-wise analysis. Notably, there was an increase in the usage of these categories after 2015, indicating more extensive research experiments during this period. By analysing the literature, we observe that ML algorithms like SVM, NB, and RF were often used for classification problems (Nair et al., 2017; Srivastava et al., 2020).

Researchers also employed these algorithms to address relevance filtering problems, utilising simple binary classifications to identify content related to a specific topic. Subsequently, this relevant data was used as input for further analysis (Khaleq and Ra, 2018; Loynes et al., 2022). ML and NN methods were primarily adopted for data classification, particularly for the detection of disaster events within the content of social media posts. These approaches have yielded promising results (Zhang et al., 2019b; Madichetty, 2020). However, it is worth noting that there was relatively less focus on analysing posts in the pre-event,

during-event, and post-event phases. Pre-event posts typically contain information related to warnings, alerts, or precursor events (Chatfield and Brajawidagda, 2013; Carley et al., 2016b).

Analysing such posts could provide valuable insights for early warning and preparedness efforts. During-event posts often contain critical information about the immediate needs of the public, such as requests for rescue, evacuation, food, or information about rescue camps. Analysing these posts can significantly aid disaster relief efforts (Jongman et al., 2015; Jitkajornwanich

et al., 2018). Post-event posts are valuable for recovery analysis, particularly in terms of damage assessment and understanding the aftermath of a disaster. Many articles in the database employed NLP, ML, or NN techniques to classify and analyse post-event data, contributing to damage assessment and recovery efforts (Shi et al., 2019; Rahmadan et al., 2020).

In our recommendations, we emphasise the importance of investigating pre-event posts, as they can provide critical information for early warning systems, helping to save lives and reduce the impact of disasters before they strike a location. Analysing

pre-event posts can contribute to better disaster preparedness and timely responses.



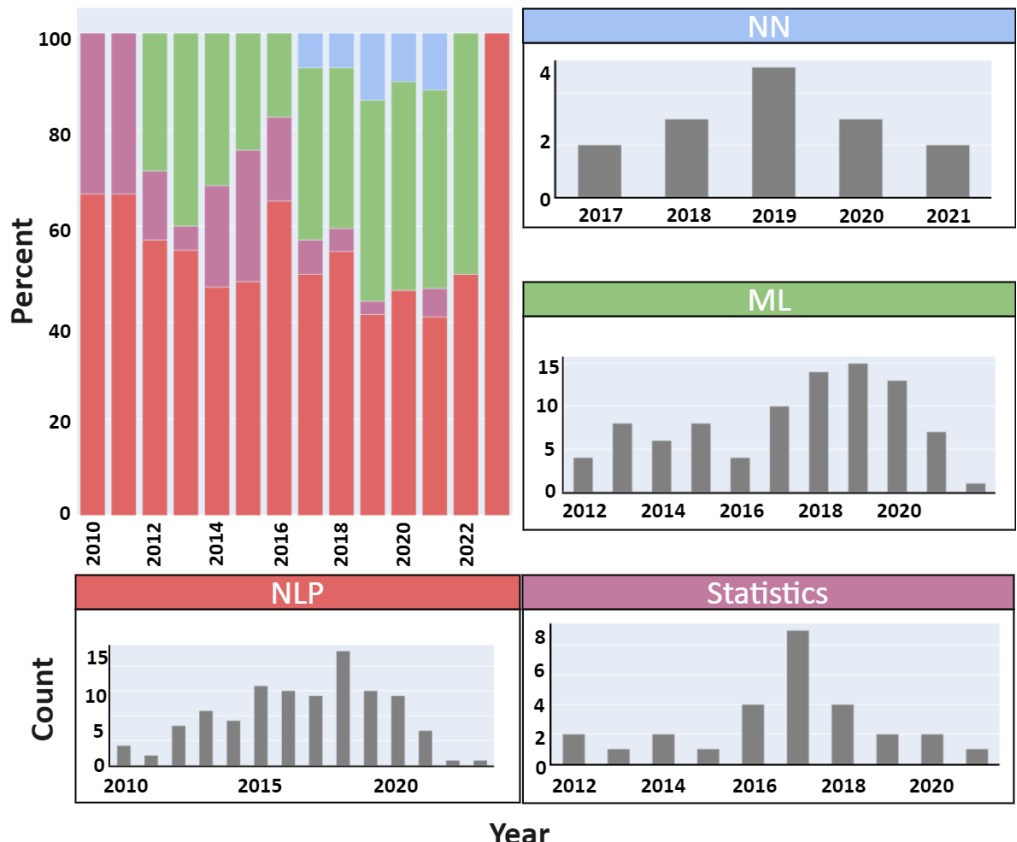

**Figure 16.** The figure shows the analysis of the methodology categories employed in the articles listed in the database - The stacked bar chart shows the percentage of 'NLP', 'ML', 'Statistics', and 'NN' categories year-wise. The sub-bar plot shows the number of articles that used a particular category year-wise.

## 5.5 Actionable Information in Database Articles

The articles categorised under the 'Disaster Management' theme was further classified into nine actionable information (AI) classes to address our research question Q2. Our goal was to analyse whether the literature provides actionable data from social media content to improve disaster management. Our analysis revealed that each article fell into at least one AI class, with AI-1,

AI-2 and AI-3 being the most commonly addressed categories in the literature (Li et al., 2018b; Wang et al., 2018). Conversely, AI-8 received the least attention, indicating that researchers have explored the identification of resources from social media posts less during event classification. This highlights the relatively limited exploration of during-event classifications in the reviewed literature (Houston et al., 2012; Kryvasheyeu et al., 2016).

AI-7 is the next category in line, and it received relatively less attention from researchers, possibly due to a lack of develop-
ment and availability of open-source platforms during the review period (LÓPEZ-MARRERO, 2010; Carley et al., 2016a). We



recommend that researchers consider creating more platforms or applications for making disaster-relevant data and real-time analysis available to the public. Such platforms can significantly contribute to rapid disaster relief, response, and recovery efforts.

AI-8, which focuses on community interaction analysis, received limited attention from researchers. Studying how communities respond and interact during and after a disaster is essential for gaining a broader perspective on disaster management. It can aid in creating effective strategies for mitigating disasters in the future (Valenzuela et al., 2017; Delilah Roque et al., 2020). We encourage researchers to delve deeper into the study of community interactions and analyse behavioural changes in pre-event, during-event, and post-event phases through social media data.

While most articles in the literature database fell into one or two AI classes, a select few managed to be categorised under five or more AI classes, achieving significant results. The articles that achieved 5 or more AIs are described in table 2. Articles achieving 5+ AI classifications focused primarily on floods and hurricanes. They consistently satisfied AI-1, AI-2, and AI-3 by collecting and filtering social media data and considering temporal and spatial analysis. Some also addressed AI-4, emphasising community engagement to enhance preparedness and response.

Table 2: Analysis of the AI categories in the literature database - Table shows the key articles that gained 5 or more AI classifications (AI-1: Data Collection; AI-2: Geolocation Detection; AI-3: Relevance Filtering; AI-4: Community Collaborations; AI-5: Disaster Trends; AI-6: Stakeholder Collaborations; AI-7: Software Development; AI-8: Resource Identification; AI-9: Community Response).

| Article | AIs | Purpose | Event Type |
|---|---|---|---|
| Fohringer et al. (2015) | 1, 2, 3, 5, 8 | Discusses a new methodology for using social media data to task the collection of remote-sensing imagery during disasters or emergencies, specifically for damage assessment of transportation infrastructure. | Flood |
| Haworth and Bruce (2015) | 1, 2, 3, 5, 7 | Proposes a method based on the "wisdom of the crowd" principle for decision support during disasters. The goal is to create a flood probability map by combining multiple observations over the affected area, each assessed for its reliability. | Flood |



Table 2: Analysis of the AI categories in the literature database - Table shows the key articles that gained 5 or more AI classifications (AI-1: Data Collection; AI-2: Geolocation Detection; AI-3: Relevance Filtering; AI-4: Community Collaborations; AI-5: Disaster Trends; AI-6: Stakeholder Collaborations; AI-7: Software Development; AI-8: Resource Identification; AI-9: Community Response).

| Article | AIs | Purpose | Event Type |
|---|---|---|---|
| Yu et al. (2018) | 1, 2, 3, 4, 6, 7, 8 | Introduces an interdisciplinary approach to disaster relief management. The proposed framework combines dynamic and static databases, including social media and authoritative data, to model rescue demand during a disaster. | Hurricane |
| Guy et al. (2010) | 1,2,3,4,7,8 | Presents an architecture for emergency situational awareness by employing natural language processing and data mining techniques to extract situational awareness information from Twitter messages generated during disasters and crises. | General Emergency |
| Chae et al. (2012) | 1,2,3,4,5 | Explores connections and patterns created by the aggregated interactions in Facebook pages during disaster responses. Analyses social roles and key players using social network analysis. | Flood |





Table 2: Analysis of the AI categories in the literature database - Table shows the key articles that gained 5 or more AI classifications (AI-1: Data Collection; AI-2: Geolocation Detection; AI-3: Relevance Filtering; AI-4: Community Collaborations; AI-5: Disaster Trends; AI-6: Stakeholder Collaborations; AI-7: Software Development; AI-8: Resource Identification; AI-9: Community Response).

| Article | AIs | Purpose | Event Type |
|---|---|---|---|
| Bala et al. (2017) | 1,2,3,5,7 | Introduces an approach to improve the identification of relevant social media messages during disasters. | Flood |
| Carley et al. (2016a) | 1,2,3,4,5,6 | Aims to categorise social media messages into different themes within different disaster phases during a disaster. | Hurricane |

Analysing the literature, we found that investigator-defined, topic-specific social media data collection strategies are valuable for gathering relevant data, but challenges related to noise and data reliability persist. We recommend researchers place greater emphasis on AI-8 and AI-9 to extract resource requirements from social media data, enabling informed resource allocation decisions during and after disasters.

### 5.6 Exclusionary Criteria - reducing noise in data

In a previous study, Aswathy et al. (2022), we conducted Twitter data collection to gather tweets related to disaster events, utilising inclusion keyword sets to identify relevant data. Upon analysing the initial data collected, it became evident that data containing the inclusion keyword set but unrelated to the disaster theme was present, contributing to the noise. A comprehensive examination of the data led us to create a set of exclusion keyword sets. A few examples may include "landslide election", "landslide victory", "flood of emotions", "landslide lyrics", "market flooded" and more. This effort resulted in the identification of approximately 56 keywords related to elections, music, life quotes, market, and emotions.

As previously mentioned, it is important to evaluate implementing exclusion criteria to ensure that no relevant data is inadvertently excluded. In our methodology, we sampled 1000 tweets and assessed the performance. Figure 17 presents the results of this evaluation, indicating the number of tweets correctly excluded and the number of relevant tweets that were potentially missed when utilising exclusionary criteria.

From figure 17 we can observe that around 80% of data was filtered correctly and around 14% of noise was collected, combining results from the 4 exclusion criteria sets. From the music exclusion criteria, we missed 20% of relevant data. This method can be employed for first-level filtering to reduce noise. The limitation of this work is that the exclusion keyword set is static



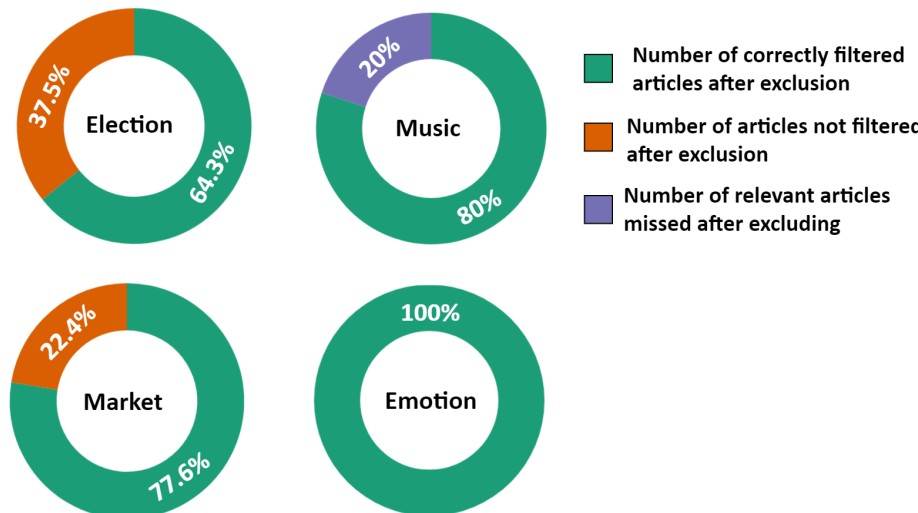

**Figure 17.** Evaluation results of the exclusionary criteria methodology: The donut charts show the percentage of tweets that were filtered from a sample data of 1000 tweets. Each sub-donut chart represents the different exclusion categories.

and needs constant maintenance when new exclusionary criteria are identified manually.

Figure 18 provides a word cloud visualisation that illustrates the impact of using exclusionary criteria on the data. As the exclusion keywords are applied, it is evident that noise is significantly reduced, and disaster-relevant data becomes more prominent

in the dataset. Our observations indicate that the utilisation of exclusionary criteria is effective in noise reduction. However, one drawback is that a thorough manual analysis is needed to initially study the data and create a keyword set that can be used to formulate an exclusionary strategy.

In disaster situations, where accuracy is paramount, ML can play a pivotal role in identifying and eliminating outliers and noise. By leveraging both basic NLP and advanced ML, researchers can aspire to achieve a comprehensive strategy for data

collection, ensuring that the information extracted from social media during crises is both accurate and actionable.

**5.7  Methodological Biases in Disaster-Related Social Media Studies in Database Articles**

This section describes seven biases within the Social Media Literature Database (Gopal et al. (2023)) articles regarding geographic, methodological, and data-related tendencies. We recognised these biases as the critical review methodology proceeded, solidified through extensive discussions among the authors. By identifying these biases, we aim to enhance the transparency of

our analysis and provide a foundation for future research.

1. *Geographic location of the case studies used in disaster-related articles*

   A notable geographic bias was found in the case studies employed by researchers in the literature, with a predominant focus on North America (see figure 9). Around 40% of the articles (60 of 154) used Hurricanes as the case study event, among which around 60% of the articles used events from North America (Kryvasheyeu et al., 2016; Mukkamala and




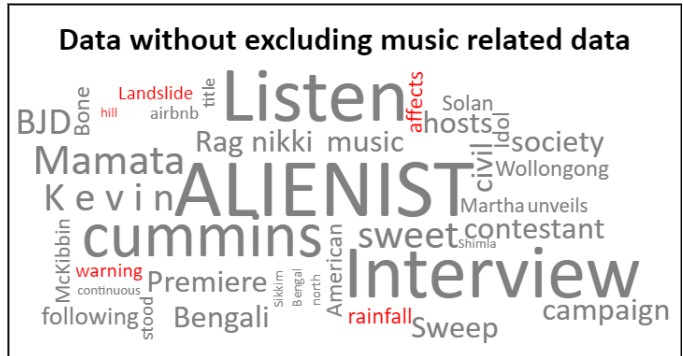

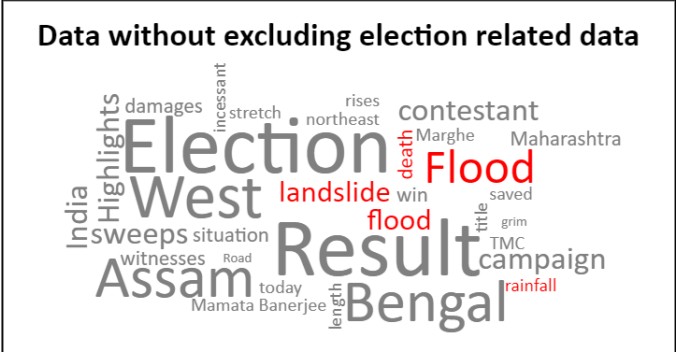

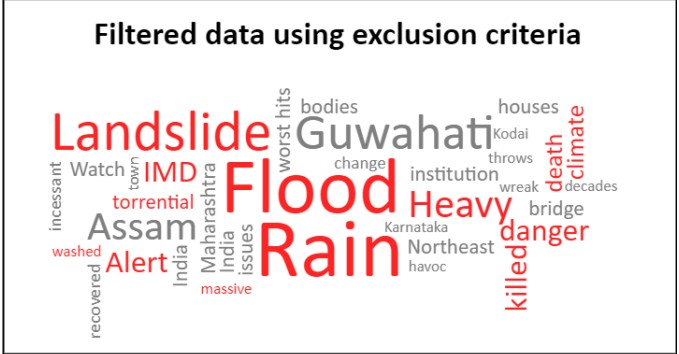

**Figure 18.** The word clouds show frequently occurring words before and after filtering the data using exclusionary criteria. The words highlighted in red are related to disasters. The size of each word represents the frequency of occurrence of a word in the sample data.



Beck, 2016; Jamali et al., 2019). This raises concerns about the generalisation of the findings in a global context.

The majority of studies exhibit a bias towards regional and national investigations, overshadowing the importance of local studies (see figure 9). One of the reasons is the availability of data is limited from a local scope when compared to a national or regional disaster event. The amount of population that uses social media platforms also varies based on the area scope. Such biases may limit the applicability of findings to specific contexts (Liu and Stevenson, 2013;

Kankanamge et al., 2020a; Li et al., 2021).

2. *External data used in disaster-related articles for methodology validation*

Various articles in the literature use external data such as EM-DAT, FEMA, USDS, AIDR, and more as supporting data to validate the methodologies employed (Spinsanti and Ostermann, 2013; Hodas et al., 2015; Liu et al., 2018). This may introduce a bias as these datasets may not comprehensively represent the effects of a disaster that occurred in a specific

region.

3. *Social media data language preference in the article*

The articles that used social media data predominantly focused on the English language, which raises a linguistic bias potentially excluding valuable insights from non-English sources. Around 6% (15 of 250) articles used a language that is regional and relevant to their respective case studies (Lee et al., 2011; Jongman et al., 2015; Radianti et al., 2016).

4. *Social Media Platform preference for data collection methodology in the articles*

A platform bias is evident, with the majority of studies drawing from Twitter data (Earle et al., 2011; Sakaki et al., 2012). This bias may limit the understanding of disaster dynamics on other social media platforms. The Twitter platform provides user-friendly API usages and sufficient meta-data making it easy for researchers to extract data (Jitkajornwanich et al., 2018). It is also important to emphasize that platforms such as Facebook and Weibo were also utilised in a few

articles in the database (Xu et al., 2016; Li et al., 2018a; Fang et al., 2019).

5. *Disaster events used in the articles for case studies*

The articles listed in the literature database predominantly explore hurricanes and floods (Le Coz et al., 2016; Madichetty, 2020), neglecting other impactful events such as pandemics, landslides, storms, and cyclones, which are few (Islam and Walkerden, 2015; Musaev et al., 2018). This may overlook crucial aspects of disaster dynamics (see section 5.2). It is

also relevant to analyse precursor events, such as heavy rain as a precursor of a flood or a landslide, which aids in early warning and mitigation.

6. *Preference of disaster management phase in the articles for case studies*

A bias emerges towards post-disaster phases such as response and recovery, with limited exploration of early warning and mitigation phases. Around 4% (7 of 157) articles experimented with early warning methodologies (Leon et al., 2018;

Wu and Cui, 2018; Kitazawa and Hale, 2021). This raises the concern about social media data availability in real-time from the social media platforms to develop solutions for early warning and mitigation.

Figure 19 shows the number of articles categorised under each disaster management phase, yearwise, and we can observe





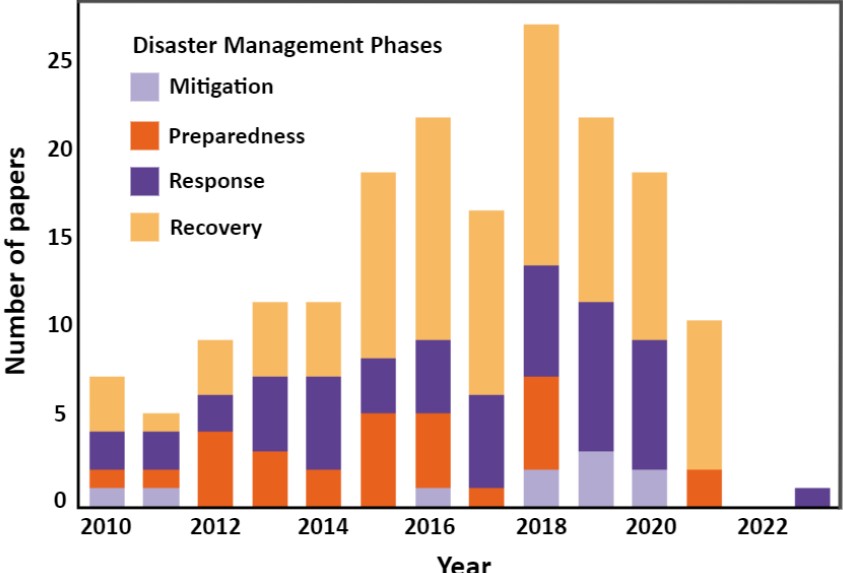

**Figure 19.** The stacked bar chart shows the number of articles listed in the literature database categorised under each disaster management phase, yearwise.

that post-disaster phases, which include response and recovery are discussed more when compared to mitigation and preparedness. We recommend the investigators develop early warning solutions using social media by analysing the precursor events of a disaster.

7. *Actionable Information the methodologies of disaster-related articles*

While researchers excel in temporal and spatial analysis (Kryvasheyeu et al., 2016; Wang et al., 2018), there is a noticeable bias with limited attention given to community interaction analysis, stakeholder engagement, and resource allocation strategies hindering a holistic approach to actionable information (see figure 13 and section 4.10). Around 24% of the articles (49 of 212) discuss methods of community and stakeholder engagement to understand the needs of the public during and post a disaster event (Chae et al., 2014; Wang et al., 2016).

This critical review has systematically examined biases within the analysed articles and the methodology employed. The identification and acknowledgment of these biases are necessary for ensuring transparency in the analysis and for future research in the field. These biases bring attention to potential limitations in the existing methodologies, emphasising the need for a more inclusive approach in future studies.

## 5.8 Best Practices of Social Media Usage for Community and Researchers

Social media has emerged over the years as a tool for information dissemination during disasters offering real-time communication aiding the public, government and non-government organisations, volunteer groups, and other actors in disaster



management (Bruns and Liang, 2012; Smith et al., 2017; Kankanamge et al., 2020b). As the public becomes increasingly
reliant on these platforms to share or acquire information, it is necessary to explore the best practices to be followed by the
community and the researchers to harness the potential of these platforms responsibly (Lin et al., 2016). Based on the literature
study, we propose guidelines for the public to share information on social media and discuss the ways to acquire disaster-
relevant information for the researchers. By implementing these best practices, communities and researchers can contribute to
a more effective response, ultimately aiding in the mitigation and recovery efforts.

The community plays a crucial role in disaster response by providing valuable information to first responders (Stephenson
et al., 2018; Kankanamge et al., 2020a). Social media platforms are widely utilised for data acquisition during disasters, but
the major challenge is to identify reliable information (Khaleq and Ra, 2018; Loynes et al., 2022). Table 3 shows a few best
practices identified from the literature that can be followed by the community to provide credible information on social media.

Table 3: The proposed best practices of social media usage for the community for effective disaster response.

| # | Best Practices | Description |
|---|---|---|
| i) | Social Media Platform Selection for Effective Disaster Communication | Leveraging location-specific popular platforms enhances reach and improves information dissemination during disasters (e.g., Twitter in the USA, Facebook in India and the UK, and Weibo in China). |
| ii) | Mitigating Rumours and Misinformation | Ensuring accuracy is crucial to prevent the spread of rumors during disaster management. Avoiding assumptions and speculations in social media sharing is vital for effective information dissemination. |
| iii) | Tagging official social media handles for effective response | Every major disaster-managing government or non-governmental organisations use social media handles to share information. During or post a disaster, the public may have water, food, shelter, and rescue requirements that need immediate attention. The public can tag these handles while sharing information which aids in informing the first responders easily. |
| iv) | Provide location information in the social media post | For first responders, acquiring accurate location information is a challenge. The public can contribute effectively by sharing the location details of the affected area by geotagging or by providing landmarks or street names for efficient response. |





Table 3: The proposed best practices of social media usage for the community for effective disaster response.

| # | Best Practices | Description |
|---|---|---|
| v) | Disaster event description in the social media post | Public-provided detailed descriptions during disasters, including emergency type, severity, and visible hazards, aid first responders in assessing the situation. Using relevant hashtags of official authorities helps consolidate information for easier tracking of updates. |
| vi) | Contribute multimedia data in the social media post | Image and video information provides a better understanding of the disaster situation. By not compromising on safety and privacy, if the public can share such data, it will assist the authorities in rapid decision-making. It also provides additional credibility to the social media posts which encourages other users to forward it further. |

As researchers increasingly turn to social media data to gain insights about disasters, it is necessary to consider a few
best practices to be followed so that data can be acquired and analysed efficiently (Branz and Brockmann, 2018; Campan et al., 2018). Drawing from our comprehensive examination of the literature and our own experience in the subject, we offer recommendations to researchers as described in table 4.

Table 4: The proposed best practices of social media usage for the investigators for effective research in the field of social media and disaster management.

| # | Best Practices | Description |
|---|---|---|
| i) | Optimise data collection with clear data requirements | Conduct thorough data requirement analysis in the initial stages to devise an efficient data collection strategy. Developing a well-defined keyword set, especially for temporal and spatial-specific data, necessitates a deep understanding of the topic of interest. |
| ii) | Look beyond metadata for precise location extraction | Pay particular attention to the content within social media posts when extracting location information. This approach may yield more accurate and contextually relevant location data compared to relying solely on metadata. |
| iii) | Validate social media data with external data sources | Detecting rumours can be challenging. The usage of valid data (such as news reports, and verified social media handles, government reports) along with social media data can be experimented with for validation. |





Table 4: The proposed best practices of social media usage for the investigators for effective research in the field of social media and disaster management.

| # | Best Practices | Description |
|---|---|---|
| iv) | Improve stakeholder engagement | Stakeholder identification and network creation are highly necessary for the effective management of disasters. Through social media, a spatial analysis may assist in identifying necessary stakeholders which can in turn help in rapid communication pre-, during, and post a disaster event. |
| v) | Language inclusivity in disaster data extraction | Be language independent – focusing only on a single language could be ineffective. The community-level public may post information in local languages which may contain relevant information. |

These recommendations can enhance the effectiveness of data collection and analysis methodologies when working with social media data for disaster management.

## 5.9 Utilizing the Social Media Literature Database: Practical Applications and Recommendations

Our Social Media Literature Database is available in the form of an Excel file that is open-access on GitHub (Gopal et al. (2023)). Upon accessing the database, users can employ various functionalities to facilitate their research. Following are a few examples:

i *Search and Filter:* Researchers can search for articles based on specific criteria such as year, keyword, or journal using the search option. Additionally, the filtering option enables users to view articles based on particular conditions (e.g., articles published in a specific year).

ii *Sort Data:* The sorting option allows users to organize the data in ascending or descending order based on parameters such as year, citations, and the number of data used.

iii *Advanced Data Extraction:* Advanced users with proficiency in Excel can utilize formulas to perform complex data extractions. For example, researchers can identify articles that utilize Natural Language Processing (NLP) as a methodology within a specified timeframe.

iv *Reuse for Review Articles:* In the last decade, various authors contributed critical and systematic reviews in the domain of social media and disaster management (Tang et al., 2021; Tsao et al., 2021; Bukar et al., 2022). Researchers interested in conducting review articles in their domain can follow the paper searching criteria and Boolean search string formation methodologies outlined in the database. This enables them to search for relevant papers and extract pertinent information for their review.



v *Usage for social media researchers:* While the columns in the database are tailored for social media relevance filtering in disaster management, researchers from the social media domain can adapt the database to their needs. By excluding irrelevant columns and focusing on relevant ones such as article source details, researchers can redefine the database for their specific domain.

vi *Usage for disaster management researchers:* Researchers in the field of disaster management can leverage the "Event" and "Case Study" columns to perform basic searching and sorting techniques. This allows for a detailed analysis of various disaster events in different years and locations.

## 6 Conclusions

In conclusion, the surge in social media data usage as a real-time information source has had a transformative impact on the field of disaster management (Valenzuela et al., 2017; Mazoyer et al., 2018). To leverage social media data usage for improving disaster management, the identification of relevant and credible information is the main priority (Schempp et al., 2019; Domala et al., 2020). Our critical review of 250 articles, spanning from 2010 to 2023, is available as a Social Media Literature Database (Gopal et al. (2023)) and has unveiled the methodologies, challenges, and actionable insights on how to harness the potential of social media data.

Our findings highlight the usage of diverse technological approaches employed by researchers over the years, mainly focusing on NLP (Houston et al., 2012; de Oliveira and Guelpeli, 2020), ML (Hodas et al., 2015; Domala et al., 2020), and statistical approaches (Middleton et al., 2013; Lu and Yuan, 2021) to address the challenges in identifying relevant and actionable information from social media to apply in the various phases of disaster management. We discussed various algorithms used over the years to collect and analyse social media data. These methodologies offer the means to identify noise which improves the data and relevance filtering (De Albuquerque et al., 2015; Jiang et al., 2022).

Our review also focused on the influence of historical disaster events on the researchers (Gupta et al., 2013; Ferris et al., 2016) and observed that major disaster events were considered case studies (see section 5.2). Such events will contain vast amounts of data which helps in gaining a wider perspective from multiple dimensions. By categorising the articles into nine actionable information classes (see section 4.3 and section 5.5), we observed the multifaceted usage of social media data in various applications. Notably, some researchers have achieved classification into multiple AI classes, as shown in table 2. This success points to the potential usage of social media data in disaster response, preparedness, and relief efforts.

The studies included in the critical review that employed a spatiotemporal analysis mostly studied hurricanes and floods (Rossi et al., 2018; Madichetty, 2020). We observed that Hurricane Sandy was one of the key events that was used as a case study by the researchers (Pourebrahim et al., 2019; Wang et al., 2019). Across the majority of the articles used in this review, Twitter was the most prevalent platform. Other platforms such as Facebook and Weibo were also used, but in limited numbers (Xu et al., 2016; Han et al., 2020).

Through this critical review, we were able to conclude that exclusionary criterias implemented using the current technologies such as NLP and ML, certainly aid in relevance filtering of social media data. Moreover, it highlights that while actionable in-



formation for disaster management can indeed be extracted from social media, there is a need for more emphasis on addressing data reliability (Bruns and Liang, 2012; Muhammad et al., 2018). A predominant challenge identified in social media usage for disaster management is the spreading of rumours and misinformation, a concern that holds critical implications in emergencies (Mendoza et al., 2010; Zhang et al., 2019b). Notably, our observation underscores a relatively limited focus on the analysis of community and stakeholder interactions, which holds significant potential as a major contribution to first responders during

disasters.

Our review also proposed best practices for the usage of social media to the community and researchers. We suggested methods of posting disaster-related content on social media to gain maximum reach and attention. We suggested including account tagging and hashtags of concerned authority accounts to receive attention. We also observed through the critical review that clarity in the post content and inclusion of multimedia improves credibility (Muhammad et al., 2018; Alam et al., 2018). We

suggested methods of extracting social media data to the researchers and the good practices to utilise them.

In essence, this review not only provides a comprehensive overview of the existing literature but also aims to contribute to future studies to explore various disciplines in leveraging social media data to fortify disaster management efforts.

## Appendix A:  Appendix A

Following is the list of acronyms and abbreviations used in this article.

| Acronym | Description |
| --- | --- |
| ANN | Artificial Neural Networks |
| BoW | Bag-of-Words |
| CNN | Convolutional Neural Network |
| DT | Decision Trees |
| En | Entropy |
| Gv | Glove |
| KNN | K Nearest Neighbours |
| LDA | Latent Dirichlet Allocation |
| LSTM | Long Short-Term Memory |
| ML | Machine Learning |
| NB | Naive Bayes |
| NER | Named Entity Recognition |
| NLP | Natural Language Processing |
| NN | Neural Networks |
| PCA | Principal Component Analysis |





| Acronym | Description |
| --- | --- |
| PoS | Part-of-Speech |
| Regex | Regular Expression |
| RF | Random Forest |
| SVM | Support Vector Machines |
| TF-IDF | Term Frequency-Inverse Document Frequency |
| UGI | User Generated Information |
| Vc | Vectorisation |
| VGI | Volunteered Geographic Information |

*Author contributions.* L.S Gopal and R. Prabha conceptualized and devised the methodology; L.S Gopal worked on the visualization and D. Malamud provided guidance. D. Malamud and H. Thirugnanam provided crucial supervision, review, and editing. M.V. Ramesh contributed significantly to the conceptualization and supervision phases. The manuscript was prepared by L.S. Gopal, integrating contributions from all co-authors.

*Competing interests.* At least one of the (co-)authors is a member of the editorial board of Natural Hazards and Earth System Sciences.

*Acknowledgements.* We would like to express our immense gratitude to our beloved Chancellor Sri Mata Amritanandamayi Devi (AMMA) for providing the motivation and inspiration for doing this research work. I would like to express my deepest gratitude to the late Dr. Rekha Prabha for her invaluable guidance and significant contributions to this work. We thank Ms. Emma Bee (Senior Geospatial Analyst, British Geological Survey) for her valuable input. This review was done as part of the Landslide multi-hazard Risk Assessment, preparedness
and Early Warning in South Asia: Integrating Meteorology, landscape, and Society (LANDSLIP) project. I would like to thank the team members, Mr. Ramesh Guntha, Mr. Sudarshan Navada, Mr. Y. V. Rayudu, Ms. Divya Pullarkatt, Ms. Aswathy A and Ms. Krishnendu K for their contributions. We also thank Mr. Subhilash Sadanandan, Mr. Sibu N., and Mr. Sravan Thampan for their valuable suggestions in graphical visualizations. Additionally, we acknowledge the use of AI tools for checking grammar, spelling, and rephrasing of sentences wherever required.



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
