# Peer review of "Review Article: Leveraging Social Media for Managing Natural Hazard Disasters: A Critical Review of Data Collection Strategies and Actionable Insights"

_EGUsphere, 2024_

## Referee Comment (RC1)

Referee report on „Review Article: Leveraging Social Media for Managing Natural Hazard Disasters: A Critical Review of Data Collection Strategies and Actionable Insights"

The authors present a literature review and analysis with an associated dataset of 250 articles, that they systematically retrieved to categorize concerning study area, event, data details, and methods. The authors emphasize that the evaluated studies consistently show that social media facilitates community interactions in crisis and that the main remaining concern is assuring accuracy by addressing the unreliability of the data. They specifically focus on actionable insights from the reviewed papers and present the results from the categorized literature with a number of well-compiled images. The insights they draw from the literature are timely and represent a novel contribution and a future reference for the field of social media usage and analysis during natural hazards.

Nevertheless, I have identified one point that requires improvement/clarification in the interpretation of their findings and a few minor suggestions for the manuscript.  Figure 3 shows the search term combinations, which were used to generate the database. each of the search strings contains a word related to the platform twitter, but no other commonly used platforms (e.g. Weibo, facebook, Instagram TikTok and more) are included in the search. Since the twitter API did provide researchers with free access, this search terminology is valid and it is very transparently reported. However, in L338-L339 and L685-690 the authors report a platform bias toward Twitter. This might be a trend in research, but I think the dataset is not suited to underline this finding. If the authors report this bias on basis of a dataset, that was filtered articles with the word "twitter", naturally the authors will retrieve more articles, that  base their analysis on data from the platform twitter and they should not report this as a representative result. I would assume that is the search term twitter was replaced with the search term "Weibo" we might have a much higher percentage of articles using this platform (although propably twitter would still be the number one). So here I would expect this limitation to be mentioned alongside the results and propably mention platform-related results with higher uncertainty.

Miner improvements I would suggest are

L52: "section 1" → "Section 1"

L97 "delves into" L310 "delving into" L500 "delve into" → this wording is overused by LLMs recently and therefore I would cange

Figure 1: Box "Research Question Identification" → Q1 is unclear to me

L163: I think here it would be good to specify what "primarily google scholar" means, you mention it later but it could be moved here to not leave the reader wondering

L176: "The may…" → "There may…"

L198: I think the commas should be outside the quotation marks

L283: Citation is all caps, please correct, also in following instances

L370/375: double association of the acronym AI for actionable insights and Artificial Intelligence is confusing, please use a different one

Figure 14: Why is some text in black and some in white? This makes the Figure more confusing.

L557 and Figure 15: The representation of quantities in lakh is an Indian numbering system, that is not commonly used in many other countries and therefore not  very suited for an international audience

---

## Referee Comment (RC2)

Thank you very much for your thorough response. With the additional analysis and proposed changes I do not have any more doubts regarding the manuscript

---

## Author Comment (AC1)

We thank the anonymous referee 1 for their valuable comments on our manuscript, 'Leveraging Social Media for Disaster Management: A Critical Review of Data Collection Strategies and Actionable Insights' (egusphere-2024-1536, submitted to NHESS). We have carefully considered all the comments and will make revisions to address your suggestions. Below, we provide detailed responses to each of Referee 1 comments (R1).

**R1 comment 1**: "*The authors present a literature review and analysis with an associated dataset of 250 articles, that they systematically retrieved to categorize concerning study area, event, data details, and methods. The authors emphasize that the evaluated studies consistently show that social media facilitates community interactions in crisis and that the main remaining concern is assuring accuracy by addressing the unreliability of the data. They specifically focus on actionable insights from the reviewed papers and present the results from the categorized literature with a number of well-compiled images. The insights they draw from the literature are timely and represent a novel contribution and a future reference for the field of social media usage and analysis during natural hazards.*"

**R1-1 reply**: We thank R1 for their summary and comments, recognising the timeliness and contribution our critical review makes.
* * *
**R1 comment 2**: "*Nevertheless, I have identified one point that requires improvement/clarification in the interpretation of their findings and a few minor suggestions for the manuscript. Figure 3 shows the search term combinations, which were used to generate the database. each of the search strings contains a word related to the platform Twitter, but no other commonly used platforms (e.g. Weibo, Facebook, Instagram TikTok, and more) are included in the search. Since the Twitter API did provide researchers with free access, this search terminology is valid and it is very transparently reported. However, in L338-L339 and L685-690 the authors report a platform bias toward Twitter. This might be a trend in research, but I think the dataset is not suited to underline this finding. If the authors report this bias on the basis of a dataset, that was filtered articles with the word "Twitter", naturally the authors will retrieve more articles, that base their analysis on data from the platform Twitter and they should not report this as a representative result. I would assume that if the search term Twitter was replaced with the search term "Weibo" we might have a much higher percentage of articles using this platform (although probably Twitter would still be the number one). So here I would expect this limitation to be mentioned alongside the results and probably mention platform-related results with higher uncertainty*"

**R1-2 reply**: We thank R1 for their comment regarding the Boolean search strings, noting that only the keyword Twitter is used and that no other commonly used platforms are included in the search. As you have mentioned in the comment, Twitter (currently X) data was more widely available to researchers than other platforms. The most popular social media platforms, such as Weibo, Facebook, Instagram, TikTok, Snapchat, YouTube, Telegram, and WhatsApp, do not make their crowdsourced data available freely to the public and hence they are not used much in social media and disaster management research.

To expand on this point, and we agree this was not clarified in our original manuscript, we will revise the manuscript to include the following clarifications:

1. Acknowledgment of only using Twitter in Search Terms

   Under section 3.2 (Paper Searching Criteria). We will mention that our search strings only used Twitter as a social media platform and that excluding other commonly used platforms such as Weibo, Facebook, Instagram, and TikTok may have omitted relevant research from these platforms. However, we will note that around 5% of the articles we found using the search strings that have just Twitter and included in our database mention in their articles the use of social media data from other platforms (discussed in sections 4.2.2 and 5.7).

2. Discussion of methodological social media biases used by researchers.

   **Table 1** below shows the usage of social media platforms (most common platforms, excluding personal chat platforms such as Whatsapp or Telegram) used in disaster management research in addition to Twitter. We reran our original search strings with the term 'Twitter' and then replaced the term "Twitter" in each search string (Q1, Q2, Q3, Q4, Q5) with six different social media platforms. The column "Results from Google Scholar" represents the number of articles filtered by Google Scholar using the search query. The column "Articles related to the social media platform" represents the number of articles that specifically use the respective platform, as determined by reading the abstract and title of each article.

The search query references conducted in Google Scholar (English, to present) are as follows:

- *Q1*: *allintitle: ("social media" OR "twitter") AND ("Disaster Response" OR "Disaster Mitigation" OR "Disaster Recovery" OR "Disaster Preparedness" OR "Disaster Monitoring")*
- *Q2*: *allintitle: "disaster management" AND ("social media" OR "twitter" OR "news" OR "crowdsourcing")*
- *Q3*: *allintitle: "data collection" AND ("disaster" OR "hazard" OR "flood" OR "landslide") AND ("social media" OR "twitter" OR "tweet")*
- *Q4*: *allintitle: ("social media" OR "twitter") AND ("disaster" OR "hazard") AND ("data" OR "filtering" OR "exclusion")*
- *Q5*: *allintitle: ("social media" OR "twitter") AND ("emergency response" OR "disaster relief")*

**Table 1.** *Comparison of X (Twitter) Google Search results using search strings Q1 to Q5 with replacing the word "Twitter" with other social media platforms (Facebook, Weibo, Instagram, TikTok, Reddit, Quora). Section A of the table are all results from Google Scholar, and Section B, the results after examining abstracts and titles for relevance. Note that the same peer-review article might appear under different rows.*

| Social Media Platform | No. of articles (a = original analyses Jan 2010 to Sep 2023; b = new analyses Jan 2010 to July 2024). | | | | | | | | | |
| --- | --- | --- | --- | --- | --- | --- | --- | --- | --- | --- |
| | A. Results from Google Scholar using search strings Q1 to Q5 which includes the social media platform | | | | | B. Articles related to the social the given media platform (after examining abstract and title) | | | | |
| | Q1 | Q2 | Q3 | Q4 | Q5 | Q1 | Q2 | Q3 | Q4 | Q5 |
| X (Twitter) (a) | 107 | 125 | 4 | 81 | 48 | 82 | 112 | 2 | 23 | 31 |
| X (Twitter) (b) | 123 | 145 | 5 | 117 | 79 | 85 | 117 | 5 | 38 | 34 |
| Facebook (b) | 122 | 135 | 3 | 88 | 68 | 8 | 3 | 0 | 2 | 4 |
| Weibo (b) | 115 | 131 | 3 | 88 | 65 | 5 | 1 | 0 | 2 | 2 |

| | | | | | | | | | | |
|---|---|---|---|---|---|---|---|---|---|---|
| Instagram (b) | 116 | 134 | 3 | 86 | 65 | 1 | 2 | 0 | 0 | 0 |
| TikTok (b) | 115 | 132 | 3 | 86 | 65 | 0 | 0 | 0 | 0 | 0 |
| Reddit (b) | 115 | 132 | 3 | 86 | 65 | 0 | 1 | 0 | 0 | 1 |
| Quora (b) | 115 | 132 | 3 | 86 | 65 | 0 | 0 | 0 | 0 | 0 |

In section 5.7 (Methodological Biases), we will therefore discuss these results and add a discussion point that elaborates on the fact that Twitter has historically provided more accessible data through its API compared to other platforms.

3. Highlighting Uncertainty in Platform-Related Results
We will revise the interpretation of our findings in L338-L339 and L685-690 to include a statement about the uncertainty associated with platform-related results. We will clarify that our search term selection influences the observed platform bias towards Twitter and may not fully represent the overall trend in disaster management research.

In L338-L339 and L685-690, we have noted that Twitter is the preferred social media platform for disaster management research primarily due to the accessibility of its data compared to other platforms. We recognize that reporting this as a bias is not entirely appropriate since our search strings specifically included the term "Twitter," which influenced the dataset composition.

We still believe it is important to discuss the platform preference bias from a broader perspective that includes the overall literature, not just our dataset. While our search strings were tailored to include "Twitter," this preference reflects a trend within the research community where Twitter data is more readily available and thus more frequently used.

We will revise the manuscript to clarify this point (section 5.7), ensuring that the platform-related results are presented with higher uncertainty and explicitly acknowledging the limitation introduced by our search criteria.
* * *
**R1 Comment 3**: "*Miner improvements I would suggest are -*
*L52: "section 1" → "Section 1";*
*L97 "delves into" L310 "delving into" L500 "delve into" → this wording is overused by LLMs recently and therefore I would change;*
*L176: "The may…" → "There may…";*
*L198: I think the commas should be outside the quotation marks;*
*L283: Citation is all caps, please correct, also in following instances;*"

**R1-3 Reply**: The changes have been made in the manuscript.
* * *
**R1 Comment 4**: "*L370/375: double association of the acronym AI for actionable insights and Artificial Intelligence is confusing, please use a different one*"

**R1-4 Reply**: A different acronym will be given for the term 'Actionable Information' in the manuscript. A few possible acronyms are 'ActInfo' and 'ActIn' which will be finalised post internal discussions with the authors.
* * *
**R1 Comment 5**: "*Figure 1: Box "Research Question Identification" → Q1 is unclear to me*"

**R1-5 Reply**: Figure 1 represents a flow diagram of our review process and the first box "Research Question Identification" explains the main 2 research questions we have focused on in our work. RQ1 states "Does exclusion criteria assist in relevance filtering?" and is elaborated in section 3.1. Exclusion criteria are NLP-based programmatic conditions that can be used to eliminate noise in data. This method is widely used and we seek to identify how the methodologies in the literature have used this method to filter data. We are trying to find out if using exclusionary criteria in data collection methodologies contributes to noise removal, which ultimately aids in rapid decision-making. We will modify the figure caption by mentioning which section to refer to (3.1) to read the explanation of the research questions.
* * *
**R1 Comment 6**: "*Figure 14: Why is some text in black and some in white? This makes the Figure more confusing.*"

**R1-6 Reply**: The colours were chosen for the text for better readability, and can be modified to reduce confusion. Either the boxes written in grey-black can be modified to white colour. If readability is affected, every box caption will be modified to grey-black.
* * *
**R1 Comment 7**: "*Figure 15 - The representation of quantities in lakh is an Indian numbering system, that is not commonly used in many other countries and therefore not very suited for an international audience.*"

**R1-7 Reply**: The quantities in the figure will be modified to millions using the conversion (1 lakh = 100,000).
* * *
Thank you for your suggestions and we hope the above-mentioned revisions will meet your expectations.

\*\*\*

---

## Author Comment (AC3)

We thank the anonymous referee 2 for their valuable comments on our manuscript, 'Leveraging Social Media for Disaster Management: A Critical Review of Data Collection Strategies and Actionable Insights' (egusphere-2024-1536, submitted to NHESS). We have carefully considered all the comments and will make revisions to address your suggestions. Below, we provide detailed responses to each of Referee 2's comments (R2).

**R2 Comment 1**: "*The manuscript presents a critical review of the use of social media data in the context of disaster management, reviewing a total of 250 articles from a range of disciplines. The paper is well-written and offers a valuable contribution to the growing literature on social media in disaster contexts. The paper would benefit from a clearer focus on its core research questions and a more concise presentation of the findings. The current version includes a significant amount of descriptive content, which could be streamlined to better highlight the most important insights*."

**R2-1 Reply**: Thank you for the summary and feedback. We truly appreciate your positive assessment of our manuscript and your recognition of its contribution to the literature on social media use in disaster management.

We also acknowledge your suggestion to sharpen the focus on the core research questions and streamline the descriptive content. Accordingly, we will revise the manuscript to present the findings more concisely and reduce sections with excessive descriptive content.
* * *
**R2 Comment 2**: "*Length and focus of the study: The manuscript is quite long (44 pages, including 19 figures and 4 tables). While comprehensive, much of the content is descriptive and does not directly address the core research questions. I recommend moving some of the less critical descriptive sections (particularly in Sections 3.3, 4.2, and 4.4) to the appendix. For me, the most insightful sections were 4.3 and the discussion in Section 5. A focus on these could improve the impact of the paper.*"

**R2-2 Reply**: Thank you for this constructive feedback regarding the length and focus of the manuscript. We appreciate your suggestion to streamline descriptive sections and highlight the most impactful insights.

Regarding Section 3.3, which describes the synthesis of research findings and details each column of the Social Media Literature Database, we agree that the full content may be too detailed for the main text. However, we believe it is important to retain a summarized version in the main manuscript to help readers understand the structure and reasoning behind the database design. We will revise this section to briefly summarize the main categories and subcategories in the paper, and will move the detailed descriptions of each subcategory to the appendix.

For Section 4.2, which presents early works, we have opted to relocate most of the content to the appendix, as suggested. However, a brief overview will be retained in the main manuscript (added to section 2 - Background) to preserve continuity and context for readers unfamiliar with the background.

With regard to Section 4.4, we believe this section is directly relevant to the paper's goals, as it provides insights into how disaster management research is distributed across themes and publication sources. It

also supports the structure of our classification system. Therefore, we have chosen to retain this section in the main manuscript but will tighten the narrative to improve conciseness and clarity.
* * *
**R2 Comment 3**: "*Clarification of research questions and key insights: The paper could more clearly articulate its central research questions and ensure that the findings directly address them. In particular, I found the second research question on the "actionable information" derived from social media more relevant and of interest to a broader audience. In contrast, the first question on "exclusion criteria in relevance filtering" is more technical and may be of limited interest to non-specialist readers.*"

**R2-3 Reply**: Thank you for highlighting the importance of clearly articulating the research questions and ensuring that the findings directly address them. We agree with your observation that Research Question 2, which focuses on "actionable information" derived from social media, has strong relevance to a broader interdisciplinary audience and have emphasized this further in our revisions.

Regarding Research Question 1 on "exclusion criteria in relevance filtering," we acknowledge that it is more technical in nature. However, we would like to emphasize that it addresses a critical aspect of working with social media data. Practically all researchers working with such data, regardless of domain, engage with some level of technical processing, such as querying APIs, parsing noisy content, and filtering for relevance. In such a vast and unstructured space, the ability to identify and exclude irrelevant data is not only technical but foundational to meaningful analysis. Exclusion using keyword-based filtering, NLP techniques, or ML methods is a common and necessary practice across many applications.

To make this more accessible, we will revise Section 3.1 to better explain the relevance and motivation behind Research Question 1 in a way that is understandable to a wider audience. We will also rewrite Section 4.8, where the results of this question are presented, with clearer framing and explanation, so its importance is evident even to those outside the technical field.
* * *
**R2 Comment 4: "***Practical examples in the introduction: The introduction would benefit from one or two concrete examples illustrating the kind of information that can be derived from social media in disaster contexts and how these tools have been used by researchers. This would help orient readers who are less familiar with the data sources or their potential applications.*"

**R2-4 Reply**: We agree that including examples of how social media data has been used in disaster contexts would greatly enhance the clarity and accessibility of the introduction, particularly for readers who may be less familiar with the data sources or their applications.

In response, we will revise the introduction to include brief practical examples. Specifically, we will mention how social media was used for real-time situational awareness during events such as Hurricane Sandy, where tweets were analyzed to detect flooded streets and power outages. We will also highlight how user-generated content from platforms like Twitter and Facebook has been used to assess infrastructure damage and coordinate relief efforts.
* * *
**R2 Comment 5: "***Keyword selection: The search strategy seems to have focused on Twitter, with no mention of other major platforms such as Facebook, Instagram, or Weibo. This is particularly surprising*

*given that some of these platforms have significantly higher user bases and broader geographic reach. They have also been extensively used in disaster contexts (e.g., Facebook). The rationale for this focus should be clearly explained, and the implications of this potential bias should be acknowledged more explicitly in both the methodology and the interpretation of findings (e.g., when the authors find that the great majority of studies in their database uses Twitter as noted in line 338 and elsewhere)."*

**R2-5 Reply:** Thank you for highlighting this important point. A similar comment was raised by Referee 1, and we have provided a detailed response and corresponding revisions to address this concern.
In summary, we have clarified in Section 3.2 (Paper Searching Criteria) that the search strategy focused on Twitter due to its relatively open API access, which has historically made it more accessible for data collection in disaster management research. We now explicitly acknowledge the implications of this platform preference in our methodology and in the interpretation of our findings (Sections 5.7). Additionally, we conducted a comparative keyword search using other platform names and included a summary table to contextualize the potential bias.
* * *
**R2 Comment 6**: *"Inclusion of studies not using social media: Section 3.3.4. describes the data used in studies distinguishing whether article have utilized social media data or not. It is unclear why studies not using social media were included in the database, given the paper's stated focus on social media. Of course, the authors could use these other articles as reference points to show the advantages or disadvantages of the use of social media, but this does not really happen in the analysis. Instead, when describing their database, the authors also refer to those articles not using social media data, which may confound the analysis."*

**R2-6 Reply**: We acknowledge the confusion this may have caused and agree that clarification is necessary.
We have included a small number of articles in the database that do not directly utilize social media data for two main reasons: (1) some of these are review or survey papers that critically discuss the role and application of social media in disaster management, even though they do not perform direct data collection or analysis; (2) others present methods for social media data collection or relevance filtering, but apply them only conceptually or use proxy data (e.g., user-generated content from official platforms), rather than directly collected social media data.
To address this, we will revise Section 3.3.4 and other relevant parts of the manuscript to clearly state the rationale behind including such studies and distinguish their role within the database.
* * *
**R2 Comment 7**: *"Stronger emphasis on key messages: While the paper shows great technical detail, the manuscript would benefit from a more focused discussion of key takeaways. Specifically, what are the major advantages, disadvantages, and challenges of using social media data in disaster contexts? These insights could be more prominently featured in the abstract, conclusion, and discussion sections (see also my comment 2 on the research question above)."*
**R2-7 Reply:** We agree that clearly emphasizing the key takeaways, advantages, and disadvantages would enhance the impact of the manuscript.

Advantages of using social media in disaster contexts include the availability of real-time, user-generated content from affected individuals and eyewitnesses, often enriched with metadata (e.g., time and location), which can significantly aid rapid response and decision-making.

Challenges include the reliability and credibility of content (e.g., misinformation, unverifiable posts), and the informal nature of language used, such as abbreviations, slang, or multilingual content, which complicates automated analysis and necessitates the use of NLP and ML methods.

Actionable Information: Our review identifies that social media data can provide geolocation detection, community collaboration signals, and disaster trends or hotspot identification. These capabilities directly support operational decision-making during and after a disaster, highlighting the potential of social media as a critical data source.

In relation to Research Question 1, we reinforce that keyword-based exclusion criteria form a technically sound starting point, especially considering that most social media APIs are built on keyword filters. However, for more advanced and adaptive filtering, we recommend NLP-based approaches and ML models, which can help reduce noise and improve the relevance of collected data.

We will revise the abstract, conclusion, and discussion sections to more directly highlight these insights of our review.
* * *
**R2 Comment 8**: "*Line 35: The sentence 'Existing literature reviews on social media data (SMD) platform evaluations, data collection tools, and analysis methods over time' is missing a verb.*"

**R2-8 Reply:** 'There are several existing literature reviews on social media data (SMD) platform evaluations, data collection tools, and analysis methods over time.' This revision corrects the grammatical issue and clearly states the existence of prior reviews.
* * *
**R2 Comment 9**: "*Line 176: Typo in "The may however be other keywords..."*"

**R2-9 Reply:** 'There may, however, be other keywords that we did not use which would have identified further relevant literature.' This revision corrects the typo.
* * *
**R2 Comment 10:** "*Line 180ff: The authors note that relevant studies may have been missed. Would it not have been advisable to revise the keyword strategy to more comprehensively capture the literature?*"

**R2-10 Reply:** We agree that refining the keyword strategy could have led to a more comprehensive capture of the literature. However, as part of our critical review methodology, we proceeded with analyzing the 250 articles identified in the first phase (Phase I) of our review, which used the Boolean search strings initially developed based on our topics of interest.

In Lines 180–185 under Section 3.2, we explicitly acknowledge this as a potential limitation and highlight it as a methodological bias. We intended to maintain transparency about the scope of our keyword strategy and its influence on the literature selection. In future studies, we plan to incorporate an iterative keyword refinement process to broaden coverage. This point will be clarified in the revised manuscript.
* * *
**R2 Comment 11:** "*Line 288: Missing word: 'Such data is considered credible when compared to the data extracted from social media platforms as it may include false information.*"

**R2-11 Reply:** The updated version now reads: "Such data is considered more credible compared to data extracted from social media platforms, which may include false or misleading information."
* * *
**R2 Comment 12:** "*Section titles: Some subsection titles could be made more meaningful and specific. For example, "4.2. Early Works" could clarify the time period; "4.2.2. Previous Works on Social Media Analytics" could clarify what is meant with "previous", etc.*"

**R2-12 Reply:** The section titled "4.2 Early Works" refers to studies published between 2010 and 2023; we will update the title to reflect this time frame explicitly. Similarly, "4.2.2 Previous Works on Social Media Analytics" also refers to early studies within the same period (2010–2023). We will revise the heading to avoid ambiguity. Additionally, we will review other section titles across the manuscript to enhance clarity.

Thank you for your suggestions, and we hope the above-mentioned revisions will meet your expectations.
* * *